# Identifying and Mitigating Errors in Gradient Aggregation of Distributed Data Parallel Training

**Zhenheng Tang** [* 1]  **Junlin Huang** [* 2]  **Zichen Tang** [* 1]  **Xueze Kang** [2]  **Yuxin Wang** [2]  **Peijie Dong** [2]  **Shaohuai Shi** [3]  **Xiaowen Chu** [2]  **Bo Li** [1]

## Abstract

Hardware-related silent data corruptions during gradient aggregation pose significant challenges to fault-tolerant distributed training, often leading to slow or failed convergence. To address this, we first mathematically formulate these errors as gradient inconsistency and theoretically analyze how they result in accumulated model divergence. Guided by this analysis, we introduce `PAFT`, a fault-tolerant distributed training system designed with dynamic and asynchronous parameter synchronization. `PAFT` comprises two core components: `PAFT-Sync`, which mitigates divergence via periodic synchronization, and `PAFT-Dyn`, which minimizes overhead through dynamic training overlap and frequency scheduling. Furthermore, the system's synchronization mechanism is optimized to support standard optimizers, including SGD, SGD momentum, and Adam. We implement `PAFT` on PyTorch Distributed, and experimental results training ResNet, GPT-2, and LLaMA-2 on 4∼32 GPUs demonstrate that it efficiently defends against aggregation errors while maintaining training performance.

## 1. Introduction

To efficiently train deep learning (DL) models (He et al., 2016) and large language models (LLMs) (Radford et al., 2018; Chung et al., 2022), high-performance and large-scale distributed training frameworks have been proposed (Rasley et al., 2020; Narayanan et al., 2021; Tang et al., 2023; 2025). Frequent system failures suspend training and require manual recovery from checkpoints, significantly

reducing system efficiency and GPU utilization (up to 43%) (Maeng et al., 2021; Wang et al., 2023b). Approximately 178,000 GPU hours were wasted during the OPT-175B training (Zhang et al., 2022) due to various failures like MPI and CUDA errors (Humbatova et al., 2020), and hardware failures such as GPU malfunctions (Hu et al., 2024), electronic breakdowns, and node failures (Wang et al., 2023b; Hu et al., 2024). Many existing studies focus on improving the robustness and efficiency of the system through fast recovery (Wang et al., 2023b; 2024; Narayanan et al., 2021) or elastic training (Thorpe et al., 2022; Harlap et al.; He et al., 2023a).

However, unlike system failures, *silent data corruption (SDC) errors* (Wang et al., 2023a; Fiala et al., 2012; Bacon, 2022; He et al., 2023b), which do not directly interrupt training, are increasingly affecting model quality and convergence. As reported in LLaMA-3 pretraining cluster and Fire-Flyer cluster, SDC errors have become the main cause of LLM convergence issues, and the secondary cost of fault tolerance during pretraining (Dubey et al., 2024; An et al., 2024), harming the reliability and efficiency of GPU clusters at extensive scale. (We provide more real-world error types and frequency during LLM pretraining in Appendix C).

In this work, we consider the errors happen during gradient aggregation (GA), which are caused by hardware failures like bit corruptions (Jeon et al., 2019; Tiwari et al., 2015; Gao et al., 2023; Hu et al., 2024) and communication noise on network links (Hu et al., 2024; Gill et al., 2011; Tan et al., 2019; Gao et al., 2023; Khan et al., 2023), as shown in Fig. 1. Specifically, the communicated messages are aggregated and broadcasted with noise, leading to different gradients on workers, which results in slow or failed convergence. To this end, we propose the following research questions.

*How do silent errors in gradient aggregation influence distributed training and how to capture and mitigate them?*

In this work, we formulate and generalize *gradient inconsistency* (in Section 2) errors, where workers obtain different noisy averaged gradients instead of the accurate averages. We then theoretically demonstrate that this gradient inconsistency leads to accumulated model divergence (in Section 3),

---

[*]Equal contribution  [1]The Hong Kong University of Science and Technology  [2]The Hong Kong University of Science and Technology (Guangzhou)  [3]Harbin Institute of Technology. Correspondence to: Xiaowen Chu <xwchu@ust.hk>, Bo Li <bli@cse.ust.hk>.

*Proceedings of the 43rd International Conference on Machine Learning*, Seoul, South Korea. PMLR 306, 2026. Copyright 2026 by the author(s).

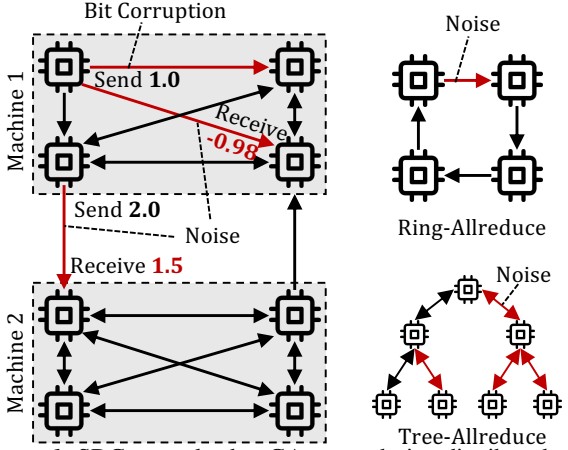

*Figure 1.* SDC errors lead to GA errors during distributed training. We provide more discussions about real-world cases in Appendix C.

resulting in failed convergence. Additionally, we quantify the convergence error theoretically concerning the degree of gradient inconsistency.

To address the GA errors at scale, we design `PAFT`, a fault-tolerant distributed training system with two components: `PAFT-Sync` and `PAFT-Dyn`. Our primary deployment target is the data-parallel (DP) dimension in hybrid parallel training, where TP/PP-partitioned model instances are replicated across workers. `PAFT-Sync` periodically synchronizes model parameters with a frequency $H$ to eliminate the model divergence. Then, `PAFT-Dyn` overlaps synchronization with the training process through asynchronous communication to save parameter synchronization overhead. To further reduce unnecessary communication costs, `PAFT-Dyn` adjusts the synchronization frequency $H$ according to the signal-to-noise ratio as observed in our theoretical convergence analysis. Our theoretical and empirical studies show that `PAFT` can alleviate accumulated model divergence, ensuring training convergence.

We implement `PAFT` on PyTorch Distributed (Ansel et al., 2024) and extend it to DeepSpeed (Rasley et al., 2020) and Megatron (Narayanan et al., 2021) for deployment on the DP dimension of real-world hybrid-parallel LLM training. We summarize our contributions as follows:

- We formulate gradient inconsistency caused by silent GA errors. We theoretically analyze how it leads to accumulated model divergence and failed convergence.

- We design `PAFT`, a fault-tolerant distributed training system to alleviate the gradient inconsistency. We theoretically prove that `PAFT-Sync` can illuminate the model divergence and ensure convergence. To reduce the extra communication overhead, we design `PAFT-Dyn` to overlap synchronization with training, and adjust the synchronization frequency with respect

to the profiled error degree based on the theoretical analysis.

- We conduct real-world experiments with 8-node GPU cluster with $4 \sim 32$ GPUs to train ResNet-18 with CIFAR-10 (Krizhevsky et al., 2010), ResNet-50 with CIFAR-100 (Krizhevsky et al.), and LLMs including GPT-2 (Radford et al., 2019) and LLaMA-2 (Touvron et al., 2023) with OpenWebText (Gokaslan et al., 2019) and Alpaca (Taori et al., 2023). We consider noises with different patterns to simulate the SDC errors with different degrees. Results show that our method can successfully mitigate these errors.

**Conflict of Interest Disclosure.** All authors are affiliated with academic institutions. This work did not use industry resources, and it does not evaluate or promote any proprietary product, system, or technology developed by an employer of any author. We use GPT-2 and LLaMA-2 only as established research models for small-scale experimental validation, and none of the authors has any employment, financial relationship, or other substantive conflict of interest with the companies that developed these models.

## 2. Preliminaries

We first present the preliminaries of distributed training, incorporating both image classification (He et al., 2016) and language modeling tasks (Radford et al., 2019). Then, we formulate the gradient inconsistency caused by the SDC errors during communication. With a model parameterized by $\theta \in \mathbb{R}^d$, and sampling data $x \sim \mathcal{D}$, the objective function is usually defined as (Bottou et al., 2016)

$$\min_\theta F(\theta) \triangleq \mathbb{E}_{x \sim \mathcal{D}} f(\theta; x), \tag{1}$$

in which the specific definition of $f(\theta; x)$ depends on the task, and it is a general formulation in many deep learning optimization problems (Dean et al., 2012). We assume that $F(\theta)$ is $L$-smooth and $\mu$-strongly convex. The convexity of $F(\theta)$ ensures that for any $\theta_1, \theta_2 \in \mathbb{R}^d$:

$$F(\theta_1) \geq F(\theta_2) + \langle \nabla F(\theta_2), \theta_1 - \theta_2 \rangle + \frac{\mu}{2}\|\theta_1 - \theta_2\|^2, \tag{2}$$

where $\mu \geq 0$. While deep learning loss landscapes are generally non-convex, assuming local convexity or strong convexity is a standard and necessary practice in optimization theory to derive convergence upper bounds and quantify the divergence caused by inconsistent gradients. For image classification, the $f(\theta; x) = l(\rho_\theta(x_i), x_o)$, where $x_i$ is the data inputs, $x_o$ the labels in the data sample, $x = (x_i, x_o)$, $\rho_\theta(x_i)$ is the output of model $\rho_\theta$, $l$ is any classification loss function, like the cross-entropy. For next-word prediction in LLMs (Radford et al., 2019; Yang et al., 2019), the

$f(\theta; x) = l(\rho_\theta(x_{1:n}), x_{n+1:N})$, where the sequence length of the $x$ is $N$. Given the seen tokens indexed by $1 : n$, the model predicts the unseen tokens indexed by $n + 1 : N$.

**Distributed SGD (DSGD).** In distributed training, multiple workers $\mathcal{M} = \{m | m = 1, 2, ..., M\}$ collaboratively optimize $\theta$. In $t$-th iteration, each worker calculates the local gradient $g_m(\theta_t^m)$. Then, the training system uses collective communication (Shi et al., 2021a; Thakur et al., 2005; Tang et al., 2020) or a parameter server (Jiang et al., 2020; Tang et al., 2020) to aggregate and broadcast the averaged gradient across workers to update model parameters $\theta$. This distributed gradient computation and model updating can be formulated as follows.

$$\bar{g}_t = \frac{1}{M} \sum_{m \in \mathcal{M}} g_t^m(\theta_t^m; x_t^m) = \frac{1}{M} \sum_{m \in \mathcal{M}} \nabla f(\theta_t^m; x_t^m), \quad (3)$$

$$\theta_{t+1}^m = \theta_t^m - \eta_t \bar{g}_t, \ x_t^m \sim \mathcal{D}_m, \quad (4)$$

where $\mathcal{D}_m$ represents dataset on worker $m$, $g_t^m(\theta_t^m; x_t^m)$ represents the local gradient of $f(\theta_t^m)$ of worker $m$ at iteration $t$, and the $\theta_t^m$ is updated with the average of local gradients $\bar{g}_t$. Normally, local dataset $\mathcal{D}_m$ has the same distribution as $\mathcal{D}$ in distributed training. We write $g_t^m(\theta_t^m; x_t^m)$ as $g_t^m$ for simplicity. Note that all models are initialized as $\theta_0$, and all workers utilize the same averaged gradient $\bar{g}_t$ to update their local models. Thus, there is $\theta_t^m = \theta_t$ during the training process.

**Errors in Distributed Averaging Gradients.** The SDC errors (Hu et al., 2024; Gao et al., 2023) in distributed training (Malcolm, 1971; Saad, 2020) actually add the noise on the estimated average gradient $\bar{g}_t$. Thus, workers finally obtain different noised gradients $\tilde{g}_t^m$ as follows.

**Definition 2.1.** *(Inconsistent Gradient). The noised averaged gradient $\tilde{g}_t^m$ is called inconsistent gradient, if there is an individual noise $\epsilon_t^m$ generated depending on $m$-th worker added on $\bar{g}_t$.*

$$\tilde{g}_t^m = \bar{g}_t + \epsilon_t^m, \ \epsilon_t^m \sim \mathcal{N}(0, \sigma^2), \quad (5)$$

in which noise $\epsilon_t^m$ is sampled from a Gaussian distribution $\mathcal{N}$ with mean of $0$ and variance of $\sigma^2$.

**Noise Degree and Patterns.** The small $\sigma^2$ can represent the small communication noise and less frequent SDC happening. On the contrary, the large $\sigma^2$ can represent the larger noise like bit corruptions (Jeon et al., 2019; Hu et al., 2024) and more frequent happening. We consider both of these two patterns in our experiments. The noises may not consistently follow a consistent pattern during training. We also consider the burst pattern of large noise (like bit corruption) that accidentally happen during training in experiments (Section 5).

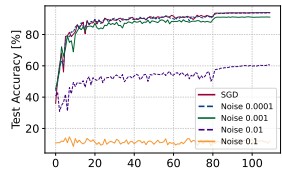
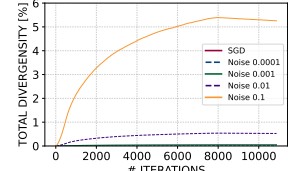

(a) Convergence Gap      (b) Model Divergence
*Figure 2.* Training ResNet-18 with gradient inconsistency on 4 workers.

## 3. Analysis of the Failed Convergence

Fig. 2(a) shows training ResNet-18 with CIFAR-10 dataset across 4 workers with and without noises $\epsilon_t^m$ with different $\sigma^2$ ranging from $0.0001 \sim 1.0$. Results show that even the small noise $0.001$ also leads to failed training convergence.

### 3.1. Accumulated Model Divergence

To understand and address this problem, we theoretically and empirically show how the gradient inconsistency (Eq. 5) leads to failed convergence. With the noised averaged gradient, the model updating process becomes from Eq. 4 as:

$$\theta_{t+1}^m = \theta_t^m - \eta_t \tilde{g}_t^m = \theta_t^m - \eta_t \bar{g}_t - \eta_t \epsilon_t^m. \quad (6)$$

At $t$-th iteration, local models $\{\theta_t^m | m \in \mathcal{M}\}$ are updated towards different directions $\tilde{g}_t^m$. Thus, this leads to diverged model parameters $\theta_t^i \neq \theta_t^j \neq \theta_t$, instead of the same $\theta_t$ in normal DSGD (Eq. 4). With training goes on, models $\theta_t^m$ gradually diverge from each other. We define the averaged model $\bar{\theta}_t = \frac{1}{M} \sum_{i=1}^M \theta_t^i$ and model divergence $\Delta_t^m = ||\bar{\theta}_{t+1} - \theta_{t+1}^m||$ to measure it. Fig. 2(b) shows the empirical accumulated model divergence during training. Larger noise (higher $\sigma^2$) introduces more divergence. This aligns with training convergence curves in Fig. 2(a), where larger $\sigma^2$ leads to a larger accuracy drop or failed convergence. Fig. 3 (b) illustrates how the noised gradients interupt convergence. Specifically, the gradients obtained on a local model $\theta^1$ result in biased gradient direction of model $\theta^2$. And local models are optimized with larger divergence.

**Lemma 3.1** (Increasing Model Divergence)**.** *With the same initial point $\theta_0^m = \theta_0$ across workers $\{m | m = 1, 2, ..., M\}$, DSGD with noise $\epsilon_t^m \sim \mathcal{N}(0, \sigma^2)$ introduces accumulated model divergence $\Delta_t^m$ during training:*

$$\mathbb{E}||\bar{\theta}_{t+1} - \theta_{t+1}^m||^2 = \frac{(M+1)\sigma^2}{M} \sum_{s=0}^{t} \eta_s^2. \quad (7)$$

**Remark.** Lemma 3.1 shows that the divergence $\Delta_t^m$ will be accumulated with the noise during training. This may lead to meaningless gradient estimation. Specifically, if the model $\theta_t^1$ is far away from the other model $\theta_t^2$, the gradient $\nabla f(\theta_t^1; x)$ has no useful descent information about the $\theta_t^1$ in the parameter space.

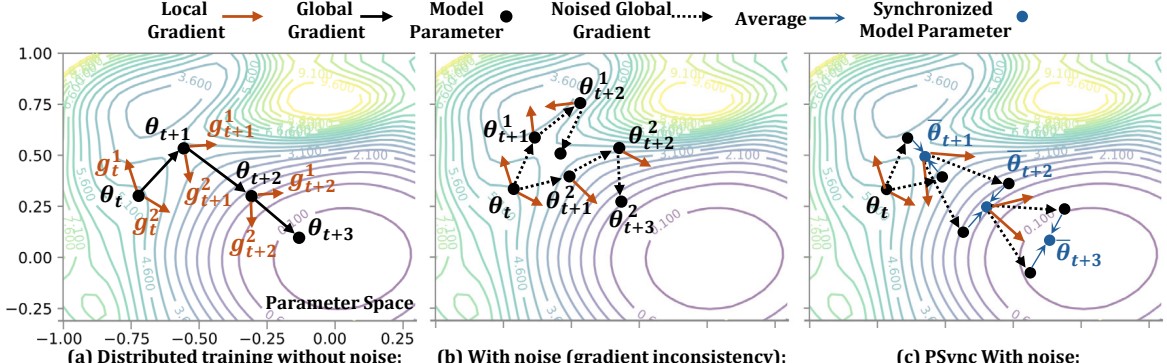

*Figure 3.* The trajectory of model parameters with training with two workers with/without noise and training with PAFT.

## 3.2. Convergence Analysis of Noised DSGD

**Assumption 3.2.** *The following assumptions are commonly used in deep learning (Bottou et al., 2016): (1) Bounded variance:* $\mathbb{E}_m ||g^m(\theta) - \nabla F^m(\theta)||^2 \le \sigma_g^2$; *(2) Bounded gradient magnitude:* $\mathbb{E}_m ||g_m^m(\theta)||^2 \le G^2$. *The* $\nabla F^m(\theta) = \mathbb{E}_i \, g^m(\theta)$ *and* $\nabla F(\theta) = 1/M \sum_{m \in \mathcal{M}} \nabla F^m(\theta)$, *and the bounded variance comes from sampling bias of the dataset on worker* $m$.

Now, we have the following theorem to show that it is difficult to tune the learning rate to have a good convergence speed.

**Theorem 3.3.** *(Convergence with noised training.) With objective function defined in Eq. 1 satisfying Assumption 3.2, and with F being convex, DSGD with noise* $\epsilon_t^m \sim \mathcal{N}(0, \sigma^2)$ *has the following convergence bound*

$$\frac{1}{T}\sum_{t=0}^{T-1} \eta_t \mathbb{E}(f(\bar{\theta}_t) - f^*) \le \underbrace{\frac{2\mathbb{E}||\bar{\theta}_0 - \theta^*||^2}{T}}_{T_1} + \underbrace{\frac{2(\sigma_g^2 + \sigma^2)}{TM}\sum_{t=0}^{T-1}\eta_t^2}_{T_2}$$
$$+ \underbrace{\frac{4L\sigma^2(M+1)}{TM}\sum_{t=0}^{T-1}\eta_t\sum_{s=0}^{t-1}\eta_s^2}_{T_3}.$$

**Remark.** In Theorem 3.3, $T_1$ and $T_2$ vanish as $T \to \infty$, but $T_3$ does not generally converge. Under a constant learning rate $\eta_t = \eta$, we have $T_3 \propto \frac{1}{T}\sum_{t=0}^{T-1}\eta\sum_{s=0}^{t-1}\eta_s^2 = \frac{\eta^3}{T}\sum_{t=0}^{T-1}t = \mathcal{O}(T)$, so the term grows linearly with training time. In principle, sufficiently fast learning-rate decay could make this accumulation convergent. However, such aggressive decay is usually impractical in modern training, where optimization commonly relies on a long constant-learning-rate phase or slowly decaying schedules such as cosine decay to maintain progress. The degenerate case $\eta_t = 0$ also yields no learning progress. Therefore, in realistic training regimes, the accumulation captured by $T_3$ remains a fundamental issue. To alleviate the model divergence in Lemma 3.1 and the accumulation captured by $T_3$, we propose PAFT in Section 4.

## 4. Periodical Parameter Synchronization

---

**Algorithm 1** Distributed training with PAFT-Sync

---

**Input:** Initialized model $\theta_0$, dataset $\mathcal{D}$, workers $\mathcal{M}$, total iteration $T$, learning rate $\eta$, synchronization frequency $H$.
**Output:** Final trained model $\theta_T$.
1: **for** $t = 1, ..., T$ **do**
2:     **for** worker $m \in \mathcal{M}$ in parallel **do**
3:         $g_t^m(\theta_t^m) = 1/B \sum_{i=1}^{B} \nabla f_{x_{t,i} \sim \mathcal{D}}(\theta_t; x_{t,i})$;
4:         $\tilde{g}_t^m = 1/M \sum_{m \in \mathcal{M}} g_t^m(\theta_t^m) + \epsilon_t^m$;    ▷
Communication
5:         $\theta_{t+1/2}^m = \theta_t^m - \eta_t \tilde{g}_t^m$;       ▷ Update model
6:         **if** $t + 1 \% H = 0$ **then**
7:             $\theta_{t+1}^m = 1/M \sum_{m \in \mathcal{M}} \theta_{t+1/2}^m$; ▷ Synchronization
8:         **else**
9:             $\theta_{t+1}^m = \theta_{t+1/2}^m$;
10: Return $\theta_T^m = \theta_T$;

---

As discussed in Section 3, the root cause of the failed convergence is the optimization of local model parameters in different directions. In this section, we begin with a straightforward but systematic solution to this issue, parameter synchronization (Section 4.1). To minimize the additional overhead, we designed PAFT-Sync to efficiently ensure training convergence (Section 4.2 and 4.3).

### 4.1. Parameter Synchronization

To eliminate the model divergence $\Delta_t^m$, one intuitive approach is to directly synchronize model parameters across workers like (Lin et al., 2018). Specifically, after updating the model at iteration $t$, workers can communicate and average their parameters $\theta_{t+1}^m$, then reload the local models as $\bar{\theta}_{t+1}$. This synchronization ensures that the model divergence $\Delta_t^m$ is eliminated, setting it to zero. However, given the model size $S_\theta$, this synchronization per iteration incurs additional communication costs amounting to $TS_\theta$, which equals the original communication costs of the gradients. Therefore, reducing the overhead of parameter synchronization is crucial.

To address this, we propose `PAFT-Sync`, as detailed in Algorithm 1 and Fig. 3 (c). In addition to standard forward and backward propagation (FP and BP), gradient averaging, and model updating, `PAFT-Sync` averages model parameters after every $H$ training iteration. The model parameters are updated as follows:

$$\theta_{t+1}^m = \begin{cases} \theta_t^m - \eta_t \tilde{g}_t^m, & \text{if } t+1\%H \neq 0 \\ \frac{1}{M} \sum_{m \in \mathcal{M}} (\theta_t^m - \eta_t \tilde{g}_t^m), & \text{if } t+1\%H = 0 \end{cases},$$
(8)

where $\tilde{g}_t^m = \bar{g}_t + \epsilon_t^m = \frac{1}{M} \sum_{m \in \mathcal{M}} g_t^m(\theta_t^m) + \epsilon_t^m$. After $H$ iterations, workers start training from the same point in the parameter space. The accumulated model divergence $\delta_t^m$ is cleared and re-accumulated at a low level, resulting in less harmful influences on gradient estimation. We theoretically and empirically demonstrate that this synchronization effectively eliminates the accumulated model divergence, thus ensuring training convergence.

**Definition 4.1.** *(gap). The gap of a set $\mathcal{A} := \{a_0, a_1, ..., a_t\}$ of $t+1$ integers, $a_i \leq a_{i+1}$ for $i = 0, ..., t-1$, is defined as $gap(\mathcal{A}) := max_{i=1,...,t}(a_i - a_{i-1})$.*

Definition 4.1 is used to generally describe the fixed and dynamic synchronization frequency in both Algorithm 1 and 2. The timestamp in sequence $\{H_t\}$ represents the synchronization point. And the $gap(\{H_t\})$ is the maximal time gap between two synchronization points.

**Lemma 4.2.** *If $gap(\mathcal{A}) \leq H$ and sequence of decreasing positive stepsizes $\{\eta_t\}_{t \geq 0}$ satisfying $\eta_t \leq 2\eta_{t+H}$ for all $t \geq 0$, then. With the same initial point $\theta_0^m = \theta_0$ across workers $\{m | m = 1, 2, ..., M\}$, DSGD with noise $\epsilon_t^m \sim \mathcal{N}(0, \sigma^2)$ introduces accumulated model divergence $\Delta_t^m$ along the training process as*

$$\mathbb{E}||\bar{\theta}_{t+1} - \theta_{t+1}^m||^2 \leq \frac{4H(M+1)\sigma^2 \eta_t^2}{M}$$
(9)

**Remark.** Lemma 4.2 shows that the model divergence is bounded with $\mathcal{O}(H\sigma^2 \eta_t^2)$. Less $H$ helps to reduce this divergence, but introduces more communication overhead. In Section 4.2 shows that `PAFT-Dyn` finds a good trade-off between the convergence and the communication in Algorithm 2.

**Theorem 4.3.** *(Convergence with noised training with* `PAFT-Sync`*.) With objective function defined in Eq. 1 satisfying Assumption 3.2, where $F$ is $L$-smooth and $\mu$-strongly convex, DSGD with* `PAFT` *(Eq. 8 or 11) and noise $\epsilon_t^m \sim \mathcal{N}(0, \sigma^2)$ satisfies*

$$\mathbb{E}f(\hat{\theta}_T) - f^* \leq \frac{\mu a^3}{2S_T} ||\theta_0 - \theta^*||^2 + \frac{4T(T+2a)(\sigma_g^2 + \sigma^2)}{\mu M S_T}$$
$$+ \frac{256T}{\mu^2 S_T} \frac{(M+1)}{M} \sigma^2 HL$$
(10)

*where $\hat{\theta}_T = \frac{1}{MS_T} \sum_{m=1}^{M} \sum_{t=0}^{T-1} w_t \theta_t^m$, for $w_t = (a+t)^2$ and $S_T = \sum_{t=0}^{T-1} w_t \geq \frac{1}{3} T^3$*

**Remark.** Theorem 4.3 shows that `PAFT` ensures the convergence of DSGD with noised gradients. And we can adjust the $H$ with respect to the noise variance $\sigma$ to trade off the convergence and communication. And Theorem 4.3 is dependent on a heterogeneous synchronization sequence $\{\mathcal{H}_t\}$ instead of a uniform sequence with the same gap $H$. Thus, it is general and can be easily extended to different algorithms that considering adjusting synchronization frequency.

**Corollary 4.4.** *Let $\hat{\theta}_T$ be defined as in Theorem 4.3, for parameter $a = \max\{16\kappa, H\}$. Then*

$$\mathbb{E}f(\hat{\theta}_T) - f^* = \mathcal{O}\left(\frac{\kappa^3 + H^3}{\mu T^3}\right)G^2 + \mathcal{O}\left(\frac{1}{\mu M T} + \frac{\kappa + H}{\mu M T^2}\right)\sigma_g^2$$
$$+ \mathcal{O}\left(\frac{(M+1)H\kappa}{\mu M T^2} + \frac{1}{\mu M T} + \frac{\kappa + H}{\mu M T^2}\right)\sigma^2$$

**Remark.** Corollary 4.4 shows that the convergence rate is the same as the SGD (Bottou et al., 2016).

---

**Algorithm 2** Distributed training with `PAFT`

**Input:** Initial model $\theta_0$, dataset $\mathcal{D}$, workers $\mathcal{M}$, total iteration $T$, learning rate $\eta$, initial detecting time gap $H_{\text{old}}$, initial synchronization sequence $\mathcal{H}_T = \{H_{\text{old}}\}$.
**Output:** Final trained model $\theta_T$.
1: **for** $t = 1, ..., T$ **do**
2:     **for** worker $m \in \mathcal{M}$ in parallel **do**
3:         $g_t^m(\theta_t^m) = \frac{1}{B} \sum_{i=1}^{B} \nabla f_{x_{t,i} \sim \mathcal{D}}(\theta_t; x_{t,i})$;
4:         $\tilde{g}_t^m = 1/M \sum_{m \in \mathcal{M}} g_t^m(\theta_t^m) + \epsilon_t^m$;
5:         **if** $t \in \mathcal{H}_T$ **then**
6:             $\theta_{t+1}^m = \theta_t^m - \eta_t \tilde{g}_t^m$;
7:             $\bar{\theta}_{t+1} = \frac{1}{M} \sum_{m \in \mathcal{M}} \theta_{t+1}^m$; (Async.)
8:         **else if** $t - 1 \in \mathcal{H}_T$ **then**
9:             Wait for $\bar{\theta}_t = 1/M \sum_{m \in \mathcal{M}} \theta_t^m$;
10:           $\sigma_{\text{est}} = ||\bar{\theta}_{p,s} - \theta_{p,s}^m||$;
11:           $H_{\text{new}} = $ All-Reduce$(||g_t^m||/\sigma_{\text{est}})$ ;
12:           Append $t + H_{\text{new}}$ in $\mathcal{H}_T$;
13:           $\theta_{t+1}^m = \bar{\theta}_t - \eta_t \tilde{g}_t^m$;
14:         **else**
15:           $\theta_{t+1}^m = \theta_t^m - \eta_t \tilde{g}_t^m$;
16: **Return** $\{\theta_T^m | m \in \mathcal{M}\}$;

---

### 4.2. Adjusting Synchronization Frequency

While the synchronization can completely address the model divergence problem, it introduces extra communication overheads due to the communication of model parameters. Through the theoretical analysis (Theorem 4.3) in Section 4.1, we adjust the synchronization frequency $H$ detected error degrees of $\epsilon$ to reduce the unnecessary communication costs.

In light of this, we propose `PAFT-Dyn` in `PAFT`, as detailed in Algorithm 2. Compared with `PAFT-Sync` (Algorithm 1), `PAFT-Dyn` detects the magnitude of error degrees

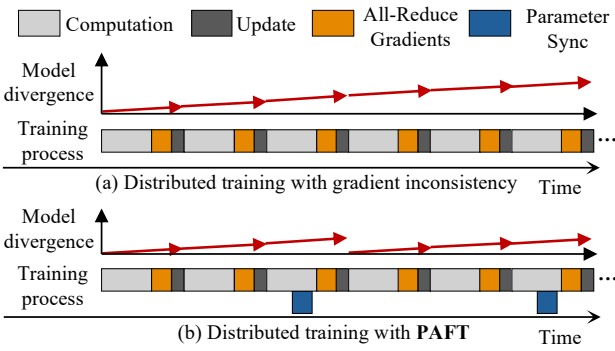

*Figure 4.* Overlapped synchronization with training.

in training (Line 10) and adjusts $H_t$ according to $\sigma_t$ and the gradient norm (Line 11) to dynamically reduce communication costs.

Then, the new parameter synchronization scheme is given as follows.

$$\theta_{t+1}^m = \begin{cases} \theta_t^m - \eta_t \tilde{g}_t^m, & \text{if } t+1 \notin \mathcal{H}_T \\ \frac{1}{M} \sum_{m \in \mathcal{M}} (\theta_t^m - \eta_t \tilde{g}_t^m), & \text{if } t+1 \in \mathcal{H}_T \end{cases}, \quad (11)$$

in which $\mathcal{H}_T$ is the sequence that indicates when to synchronize parameters.

**Estimating Error Degree.** The naive error detection method is directly computing the average of the gradients $1/M \sum_{m \in \mathcal{M}} g_t^m(\theta_t^m)$ and compare it with $\tilde{g}_t^m$ to estimate the noise degree of $\epsilon_t^m$, which introduces extra communication costs equal to synchronization. To this end, we estimate the error degree through the accumulated model divergence $\Delta_t^m$ to reduce the communication costs, as the $\Delta_t^m$ takes historical error information and need not be communicated at each iteration. According to Eq. 7 in Lemma 3.1, we can directly compute the accumulated model divergence $\Delta_t^m$ (Line 22 in Algorithm 2).

**Adjusting Synchronization Frequency.** Observing the convergence rate in Theorem 4.3, the intuitive way to adjust $H$ is set $H = \lceil 1/\sigma^2 \rceil$, thus the third term in the convergence bound (Eq. 10) becomes as $\mathcal{O}(T(M+1)L/(MS_T))$. However, this too less $H$ actually is set too small and, because the dominant bound becomes as the second term as $\mathcal{O}(2T(T+2a)(\sigma_g^2 + \sigma^2)/(MS_T))$ and cannot be reduced by smaller $H$. Thus, we can set the $H = \sigma_g/\sigma$. Now, the second term and the third term in Eq. 10 is balanced. Note that the $H = \|g_{t,p_{\max}}^m\|/\sigma_{\max}$ also represents the signal-to-noise ratio (SNR) that is widely used in many methods to adjust hyper-parameters (Qiao et al., 2021).

### 4.3. Overlapping Synchronization with Training

Furthermore, synchronization after some training iterations still requires communication. To further reduce this communication cost, we overlap synchronization with the normal backward propagation process using asynchronous communication. The timeline of this overlapped communication is shown in Fig. 4.

As detailed in Algorithm 2, if the current round requires synchronization, the model averaging process is initiated without waiting (Line 7). In the next round, the model averaging can be overlapped with the forward and backward propagation processes. During model updating, workers wait for the previous round's synchronization to be completed. The new model parameters are then updated using the averaged model and the new gradients. Note that this approach introduces a trade-off, where we trade precise gradient estimation for the benefit of overlapping communication. We show the empirical effect on eliminating the model divergence in Appendix E.

### 4.4. Extension to Other Optimizers

The analysis in Seciton 3 is mainly built on the SGD, while the most of current DL models and LLMs are optimized with SGD momentum and Adam (Kingma & Ba, 2015). However, in the noised distributed training, the intrinsic characteristics of these optimizers are similar to the SGD. Specifically, the inconsistent gradients $gtil_t^m$ also lead to diverge updating directions of the model parameters, and the accumulated model divergence. Differently, the SGD momentum and Adam introduce extra terms including the momentum and precondition, which are updated according to the gradients. Thus, there is divergence existing in these extra terms. However, the divergence on them may not be accumulated as the model parameters as they are updated with moving averaging. Nevertheless, we can consider to synchronize these extra terms with the model parameters to ensure the convergence of the model. To this end, we provides results of synchronizing the momentum and precondition in Appendix E.

## 5. Experimental Studies

In this section, we conduct experiments on distributed training with varying degrees of noise. We compare basic distributed training without gradient inconsistency (Oracle), distributed training with gradient inconsistency (Noised), PAFT-Sync with different $H$ values, and PAFT.

**Cluster Configuration.** We have two testbeds including an 8-node GPU cluster, each of which installs 4 Nvidia RTX2080Ti GPU connected with PCIe3.0x16 with 10Gbps bandwidth, and a single GPU machine equipped with 8 Nvidia A6000 GPUs.

**DL Models and Datasets.** We train ResNet-18 (He et al., 2016) with CIFAR-10 (Krizhevsky et al., 2010), ResNet-50 (He et al., 2016) with CIFAR-100 with 120 epochs, and GPT-2 Small (Radford et al., 2019) with 124M parameters

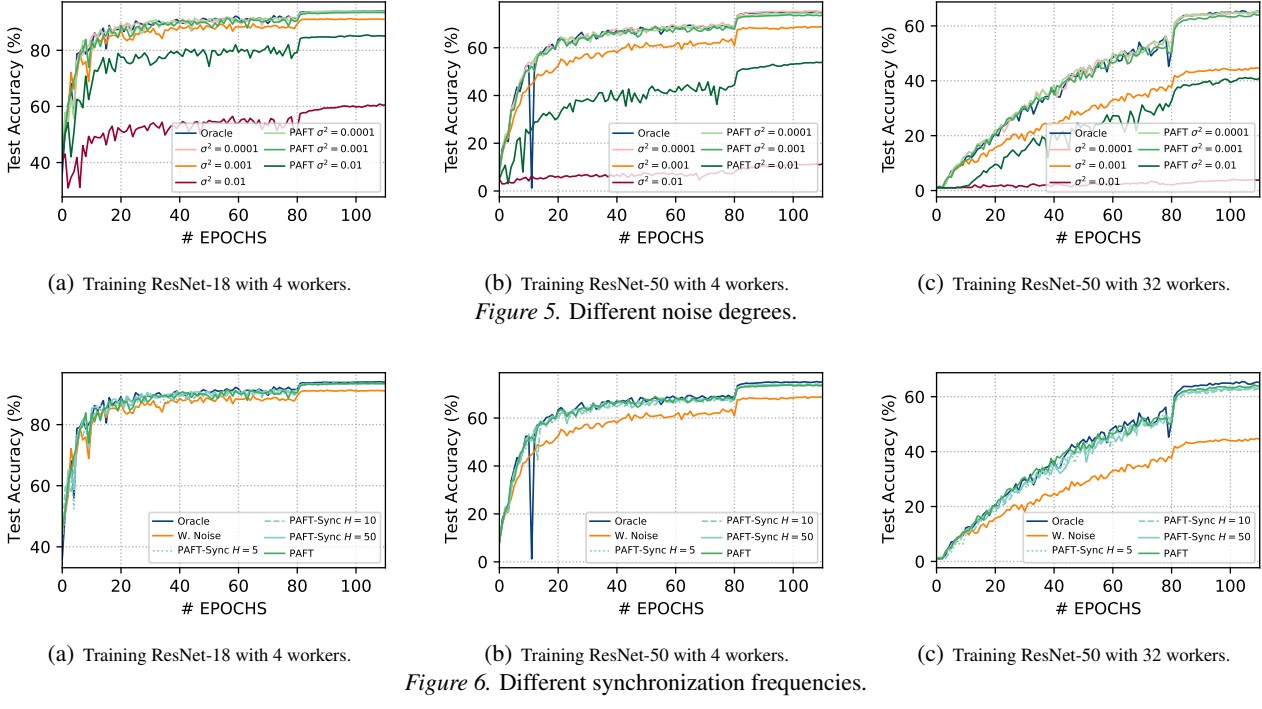

(a) Training ResNet-18 with 4 workers.     (b) Training ResNet-50 with 4 workers.     (c) Training ResNet-50 with 32 workers.

*Figure 5.* Different noise degrees.

(a) Training ResNet-18 with 4 workers.     (b) Training ResNet-50 with 4 workers.     (c) Training ResNet-50 with 32 workers.

*Figure 6.* Different synchronization frequencies.

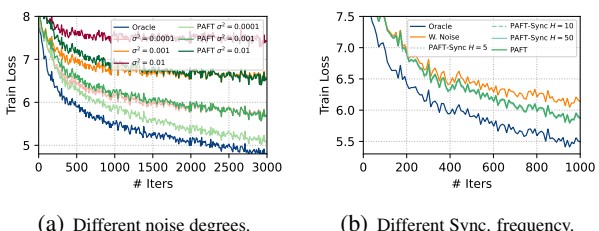

(a) Different noise degrees.     (b) Different Sync. frequency.

*Figure 7.* Training GPT-2 with OpenWebText.

on OpenWebText (Gokaslan et al., 2019) with 3K iterations. We also finetune pretrained LLaMA-2 (Touvron et al., 2023) and GPT-2 on Alpaca (Taori et al., 2023) using LoRA (Hu et al., 2021) with 1 epoch. ResNet-18 and ResNet-50 are optimized with SGDm (Bottou et al., 2016) with learning rate of 0.1 and momentum of 0.9. GPT-2 is trained with Adam (Kingma & Ba, 2015) with learning rate of 0.001, $\beta_1$ as 0.9 and $\beta_2$ as 0.99.

**Simulation of Gradient Inconsistency.** We simulate the noise with different degrees by adjusting $\sigma$ with range $\{0.0001, 0.001, 0.01, 0.1\}$. The small noise degree $\{0.0001, 0.001\}$ can represent the small communication noises. While the larger noise $\{0.01, 0.1\}$ can simulate the bit corruptions or the large communication noise, which appears less during training. This simulation is motivated by real industrial observations. In the LLaMA-3 405B pre-training report, about 78% of unexpected interruptions were attributed to confirmed or suspected hardware issues, including GPUs, GPU memory, and network components

(Appendix Table 2) (Dubey et al., 2024). In the Fire-Flyer HPC report, GPU Xid errors such as NVLink errors and ECC-related failures were observed at scale, and some of them led to gradnorm spikes, loss explosions, and even non-convergence (Appendix Tables 3 and 4) (An et al., 2024). Therefore, we use both small continuous noise and occasional large burst noise to emulate the combination of low-degree silent corruption and sporadic high-magnitude faults seen in practice.

## 5.1. Main Results

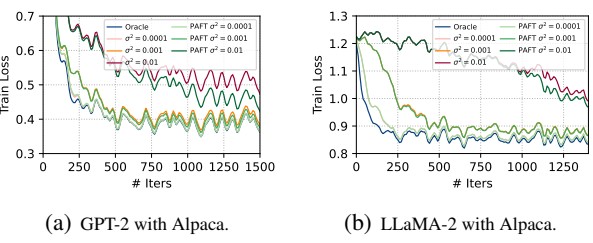

(a) GPT-2 with Alpaca.     (b) LLaMA-2 with Alpaca.

*Figure 8.* Finetuning LLMs with different noise degrees.

Fig. 5(a) and 5(b) show convergence of noised distributed training on ResNet-18 and ResNet-50 with 4 workers. Fig. 5(c) show training resnet-50 of noised distributed training with 32 workers. All results show that as noise degree increases, the accuracy of model declines correspondingly. While PAFT can successfully illuminate the small noise influence and mitigate the large noise influence.

The results in all figures show that the PAFT can success-

fully defend against noise and improve the convergence of noised training when $\sigma^2 = 0.0001$ or $0.001$. Note that there is still gap between the normal training (Oracle) and `PAFT` when $\sigma^2 \geq 0.01$. The reason is that `PAFT` is designed to correct worker-specific gradient inconsistency by removing replica drift after different workers receive different corrupted gradients. When the corruption becomes common-mode and all workers move in the same but wrong descent direction, parameter synchronization alone cannot fully recover that direction. This limitation is inherent to the fault model and motivates combining `PAFT` with orthogonal defenses such as gradient clipping or robust aggregation (Guerraoui et al., 2024).

**Training and Finetuning LLMs.** Fig. 7, 8(a) and 8(b) show the loss curves of pretraining and fine-tuning LLMs. The results show that the `PAFT` can successfully defend against noise and improve the convergence. While the model size increases from ResNets to LLMs like GPT-2 and LLaMA-2, the `PAFT` can significantly improve than baselines. When the noise degree $\sigma^2 = 0.0001$ or $0.001$, the `PAFT` can almost ensure the convergence as similar to the training without noise. While for the larger noise $\sigma^2 = 0.01$, the `PAFT` can improve the convergence compared with the noised training. The exiting performance gap between `PAFT` and the normal training without noise comes from the noisy gradient itself, which leads to an incorrect updating direction. Future works should consider combining both synchronization and voting mechanisms like the Byzantine Fault-tolerance problem (Guerraoui et al., 2024) to address this problem.

**Accidental Large Noise.** We simulate accidental large noise, like bit corruptions. Specifically, in each round, the noise is sampled from $\mathcal{N}(0, 0.0001)$ to simulate normal small noises. After each 500 iterations, the noise is sampled from a $\mathcal{N}(0, 0.1)$ or $\mathcal{N}(0, 1.0)$ as simulated accidental large noise.

The Fig. 9(a) shows training with large noise sampled from $\mathcal{N}(0, 0.1)$ while Fig. 9(b) shows $\mathcal{N}(0, 1.0)$. The convergence curves clearly demonstrate the influence of this accidental noise. In each iteration that the noise happens, the test accuracy instantly drops a lot and is pulled back by `PAFT` from the valley. However, for a large noise with variance of $1.0$, it is hard to pull it back. Interestingly, we observe that the learning rate decay at the late stage helps the model defend against the noise. Less learning rate results in less model update and divergence, which aligns with our theoretical analysis (Lemma 3.1 and Theorem 3.3).

**Wall-clock Iteration Time** We provide a comparison of the average iteration wall-clock time (in seconds) during the training of the ResNet-50 model, using different numbers of workers ranging from $4 \sim 32$ in Table 1. By dynamic adjusted synchronization frequency and overlapped commu-

Table 1. Average iteration wall-clock time (seconds) of training ResNet-50.

| # of workers | 4 | 8 | 16 | 32 |
|---|---|---|---|---|
| DSGD | 0.201 | 0.212 | 0.228 | 0.333 |
| PAFT-Sync | 0.243 | 0.254 | 0.276 | 0.411 |
| PAFT | 0.237 | 0.244 | 0.253 | 0.373 |

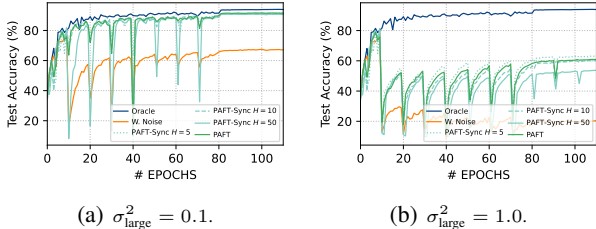

(a) $\sigma^2_{\text{large}} = 0.1$.  (b) $\sigma^2_{\text{large}} = 1.0$.

*Figure 9.* Training ResNet-18 with accidental large noise.

nication, the `PAFT` reduces the extra cost than `PAFT-Sync` for around up to 11.0% efficiency improvement for 32 workers. And the extra cost of `PAFT` than DSGD is around 18.9% for 32 workers. For more workers, `PAFT-Sync` shows better improvement, which means the good scalability of `PAFT-Sync`.

**Eliminating Model Divergence.** To directly visualize how `PAFT` works, Fig. 10 compares the convergence and model-divergence trajectories under noised DSGD and `PAFT`. Under noised DSGD, the divergence between local replicas accumulates throughout training and is accompanied by degraded test accuracy. In contrast, `PAFT` periodically pulls the replicas back to a consistent point in the parameter space, preventing sustained divergence growth and stabilizing convergence. This result empirically supports Lemma 3.1 and Theorem 4.3: parameter synchronization is effective because it erases accumulated replica drift before it dominates the optimization process.

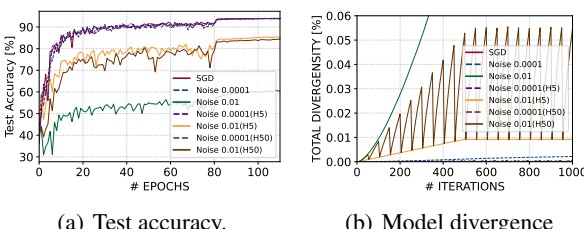

(a) Test accuracy.  (b) Model divergence

*Figure 10.* Training ResNet-18 with 4 workers: `PAFT` periodically eliminates accumulated model divergence and improves convergence under noised training.

## 6. Related Works

We provide a concise literature review here due to the limited space. We provide detailed related works in Appendix B.

**Parallelism at Scale** Distributed LM training relies on hybrid parallelism, including DP, TP, and PP (Narayanan et al.,

2021; Zhu et al., 2025). In modern systems, DP remains the key dimension for replica-based scaling and throughput improvement (Krizhevsky et al., 2017; Tang et al., 2020; 2025): a TP+PP-partitioned model instance is often replicated across multiple DP groups to increase effective batch size and cluster utilization. TP and PP, in contrast, mainly address single-device memory constraints and pipeline efficiency (Narayanan et al., 2021; 2019; Rasley et al., 2020). This distinction matters for our setting. PAFT targets GA errors on the DP dimension, where corrupted aggregation or broadcast can create replica inconsistency across workers. By comparison, communication issues in TP or PP are closer to tensor-transfer or computational corruption problems, which are outside the scope of this paper. Therefore, although pure DP alone is no longer sufficient for frontier-scale models, the DP dimension remains a critical component of modern hybrid-parallel training and is the natural deployment target of PAFT.

**Safety and Reliability** Existing reliability work mainly addresses active failures such as worker crashes through checkpointing and elastic recovery (Wang et al., 2023b; 2024; Narayanan et al., 2021; Thorpe et al., 2022; Harlap et al.). Another line studies adversarial or Byzantine settings using clipping or robust aggregation (El-Mhamdi et al., 2020; Damaskinos et al., 2018; Guerraoui et al., 2024). In contrast, GA errors in our setting are silent, unintentional, and hardware/communication induced, making them harder to detect and compensate.

**Asynchronous Optimizations** Many systems use asynchronous techniques to hide communication and storage overheads, including asynchronous optimization and checkpoint-related communication overlap (Tsitsiklis et al., 1986; Zheng et al., 2017; Mohan et al.; team, 2022; Shi et al., 2021b). Our design is related in spirit, but the goal is different: PAFT-Dyn uses asynchronous parameter synchronization specifically to reduce the overhead of fault-tolerant correction rather than to relax the synchronization semantics of standard training.

**Fault Tolerance in Federated Learning** Fault tolerance is also studied in federated learning, where client unreliability, secure aggregation under dropouts, and Byzantine-resilient optimization are central concerns (Huang et al., 2023; Mansouri et al., 2022; Fan et al., 2021; Cao et al., 2021). These works show that periodic aggregation and redundancy can improve robustness under heterogeneous failures. However, their setting differs from ours because we study replica drift caused by silent GA corruption inside distributed data-parallel training systems.

## 7. Limitations

**Compatibility with Sharded Training.** PAFT is best suited to DDP and the DP dimension of hybrid parallel training, where full parameter replicas naturally exist across workers. It is not directly compatible with purely sharded regimes such as FSDP (Paszke et al., 2017) or ZeRO-style execution, because these systems remove the parameter redundancy that PAFT relies on. A native FSDP integration would require shard-wise divergence tracking, periodic all-gather based verification to temporarily reconstruct parameters, and synchronization of sharded optimizer states, which would partially offset the memory-saving purpose of sharding.

**Synchronization Reliability and Fault Scope.** We assume the infrequent parameter-synchronization phase is executed with stronger reliability than per-iteration gradient aggregation. In practice, gradient aggregation is latency critical and typically uses high-throughput communication paths with limited end-to-end verification, whereas PAFT synchronization happens only every $H$ iterations and can therefore afford stronger protections such as TCP transport, end-to-end checksums, or software hash verification. This design directly addresses worker-specific replica drift, but it does not by itself correct common-mode corruptions shared by all workers.

## 8. Conclusion

In this paper, we address Gradient Aggregation (GA) errors in distributed training caused by hardware corruptions and communication noise. We provide a mathematical formulation of gradient inconsistency and theoretically analyze its impact on model divergence. To mitigate these risks, we introduce PAFT, which synergistically integrates PAFT-Sync for periodic parameter synchronization and PAFT-Dyn for dynamic frequency adjustment based on error profiles.

Overall, our work offers a scalable and efficient framework to enhance the reliability of large-scale distributed training systems against hardware-induced stochastic errors. More broadly, our results suggest that low-frequency, reliability-enhanced parameter synchronization is a practical systems lever for controlling silent replica drift in modern hybrid-parallel training. For future work, it is promising to validate this framework under richer real-world fault traces collected from production clusters, rather than relying only on simulated perturbations. Another important direction is to combine PAFT with orthogonal defenses such as gradient clipping or robust aggregation to handle common-mode corruptions in addition to worker-specific inconsistency.

## Impact Statement

We do not anticipate any specific negative ethical consequences, our work offers significant positive impacts in the context of large-scale distributed training. By identifying and mitigating silent data corruptions caused by hardware faults, our proposed system, `PAFT`, enhances the reliability and trustworthiness of the infrastructure underlying foundation models, ensuring trained models remain mathematically consistent with their design objectives. Furthermore, by effectively detecting and mitigating these errors on-the-fly without requiring frequent restarts, our approach substantially improves energy efficiency, reducing the substantial wasted GPU hours and carbon emissions associated with training failures. Finally, this work lowers the barrier to entry for large-scale training by providing a software-level defense against hardware inconsistencies, enabling researchers to utilize cost-effective or less stable commodity clusters effectively.

## Acknowledgement

This work was partially supported by National Natural Science Foundation of China under Grant No. 62272122, and Hong Kong CRF grants under Grant No. C7004-22G and C6015-23G.

The research was supported in part by an NSFC grant 62432008, RGC RIF grant R6021-20, an RGC TRS grant T43-513/23N-2, RGC CRF grants C7004-22G, C1029-22G and C6015-23G, NSFC/RGC grant CRS_HKUST601/24 and RGC GRF grants 16207922, 16207423 and 16203824.

This work was partially supported by the National Natural Science Foundation of China (NSFC) under Grant No. 62302123 and the Shenzhen Science and Technology Program under Grant No. KJZD20240903104103005, Grant No. KJZD20230923114213027 and Grant No. KJZD20230923115113026.

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

# Appendix

## A. The Use of Large Language Models

We used LLMs solely for grammar and wording improvements. It did not generate ideas, analyses, or results. No additional or undisclosed LLM use occurred.

## B. More Related Works

### B.1. Parallelism at Scale

Distributed large model (LM) training (Narayanan et al., 2021; Zhu et al., 2025) employs hybrid parallelism techniques, including data parallelism, tensor model parallelism, and pipeline parallelism. Recent frameworks have moved towards co-optimizing these dimensions with memory constraints to maximize training throughput (Zhu et al., 2025; Qi et al., 2025).

**Data parallelism (DP)** (Krizhevsky et al., 2017; Chen et al., 2016; Cui et al., 2016; Zhang et al., 2017; Tang et al., 2020; 2022; 2025), which replicates models for parallel training, is central in hybrid parallelism. It scales the training effectively by increasing the batch size to accelerate model convergence. However, DP is limited by memory capacity and communication overheads, especially for large-scale LM training. Recent work has proposed adaptive batch size schedules to further optimize DP efficiency while maintaining convergence guarantees (Lau et al., 2025). This paper focuses on the GA errors in DP training, a critical subset of distributed training faults recently categorized in large-scale empirical studies (Ma et al., 2025).

**Tensor model parallelism (TP)** (Or et al., 2020; Narayanan et al., 2021) complements DP by addressing memory limitations when models exceed a single device's memory capacity. `PAFT` tackles GA errors and has been generalized to hybrid parallel training frameworks like DeepSpeed (Rasley et al., 2020) and Megatron (Narayanan et al., 2021) towards large-scale LM training. The TP training may also have communication errors, specifically related to the frequent synchronization required in asymmetric heterogeneous environments (Tae Kim et al., 2025). These errors are out of the scope of this paper. And the communication errors in concatenating tensors in TP are more like the computational SDC errors, which is different from the GA errors in DP.

**Pipeline parallelism (PP)** (Narayanan et al., 2019; Rasley et al., 2020; Tang et al., 2023) splits the whole model into different stages and processes them in a pipelined manner. The PP can reduce the memory consumption and communication overheads. Recent advances like Seq1F1B (Sun et al., 2024) and synergistic scheduling (Qi et al., 2025) further reduce pipeline bubbles and memory footprints for long-context training. The communication errors in PP are more like the quantization or compression errors, which are different from the GA errors, either.

### B.2. Safety and Reliability of Distributed Training

**Active Failures.** Many studies focus on system reliability concerning node failures, which may directly interrupt training processes. These studies propose fault-tolerant mechanisms using checkpointing (Wang et al., 2023b; 2024; Narayanan et al., 2021) and elasticity (Thorpe et al., 2022; Harlap et al.; He et al., 2023a) optimizations for rapid recovery. These optimizations enhance system robustness and enable quick restarts.

**Silent Failures.** There are other soft failures like the communication noise happen in GA, or the workers upload the wrong gradients to the server. The typical methods to handle these failures include gradient clip, or considering them as the Byzantine faults by malicious node behavior (El-Mhamdi et al., 2020; Damaskinos et al., 2018; Guerraoui et al., 2024). However, the silent errors in GA errors in the scope of this paper, arise from unintentional issues like hardware errors or communication errors, leading to inaccuracies in gradient updates. And we mainly focus on the GA errors happen during broadcasting in DP training, which is different from the other types of soft failures.

### B.3. Asynchronous Optimizations

To accelerate distributed training, asynchronous optimization techniques have been proposed to reduce the synchronization overheads (Tsitsiklis et al., 1986; Zheng et al., 2017; Damaskinos et al., 2018). These techniques allow workers to update model parameters independently, reducing the waiting time for synchronization. To consider accelerating synchronizing checkpoints, many works utilize the asynchronous and heterogeneous capabilities of hardware resources for parallel processing of different tasks. For example, in checkpointing optimizations, asynchronous parameter snapshotting can

compete for memory bandwidth with training processes, potentially slowing down the training speed (Mohan et al.; team, 2022; Wang et al., 2024). Additionally, inter-node communications asynchronous to training can introduce communication overheads (Shi et al., 2020; 2021b). In PAFT-Sync, we also observe unavoidable asynchronous overheads during training. However, the dynamic synchronization frequency effectively reduces the overall asynchronous overhead in the fault-tolerant system.

### B.4. Fault Tolerance in Federated Learning

Some works investigate fault tolerance of federated learning (FL). Considering the impact of unreliable devices (e.g., dropouts, misconfigurations, poor data quality) on FL performance, especially in rural environments with limited clients, how infrastructure-level errors (e.g., unstable power, network issues) and ML-specific inconsistencies (e.g., misconfigured hyperparameters, low-quality data) affect FL model accuracy is not explored (Huang et al., 2023). It is found that FedAVG can perform well even with unreliable clients (Huang et al., 2023). This shows that the parameter synchronization might be a useful tool in different kinds of fault-tolerance scenarios.

Some works (Mansouri et al., 2022) propose a new secure and fault-tolerant aggregation scheme that can recover from client failures. Their solution is based on a threshold variant of the Joye-Libert secure aggregation scheme, combined with decentralized key management and input encoding. The scheme allows a set of available clients to compute the encryption of a zero value on behalf of missing clients, enabling the aggregator to correctly aggregate the inputs of the online clients.

And some work (Fan et al., 2021) proposes a novel FRL framework, Federated Policy Gradient with Byzantine Resilience (FedPG-BR), which ensures convergence and tolerates a certain percentage of faulty agents. The theoretical analysis of FedPG-BR demonstrates its improved sample efficiency with more agents and its resilience to Byzantine faults. The key idea is to design a gradient-based Byzantine filter on top of a variance-reduced federated policy gradient framework. Some works improve the homomorphic encryption considering the fault-tolerance (Zhang et al., 2024).

Byzantine fault-tolerant FL is a mainly focused direction (Liu et al., 2021; TANG et al., 2026). The redundancies in FL agents' cost functions are necessary and sufficient to ensure Byzantine resilience. Using a root dataset by the service provider (Cao et al., 2021) helps to detect the malicious attacks.

| Component | Category | Interruption Count | % of Interruptions |
|---|---|---|---|
| Faulty GPU | GPU | 148 | 30.1% |
| GPU HBM3 Memory | GPU | 72 | 17.2% |
| Software Bug | Dependency | 54 | 12.9% |
| Network Switch/Cable | Network | 35 | 8.4% |
| Host Maintenance | Unplanned Maintenance | 32 | 7.6% |
| GPU SRAM Memory | GPU | 19 | 4.5% |
| GPU System Processor | GPU | 17 | 4.1% |
| NIC | Host | 7 | 1.7% |
| NCCL Watchdog Timeouts | Unknown | 7 | 1.7% |
| Silent Data Corruption | GPU | 6 | 1.4% |
| GPU Thermal Interface + Sensor | GPU | 6 | 1.4% |
| SSD | Host | 3 | 0.7% |
| Power Supply | Host | 3 | 0.7% |
| Server Chassis | Host | 2 | 0.5% |
| IO Expansion Board | Host | 2 | 0.5% |
| Dependency | Dependency | 2 | 0.5% |
| CPU | Host | 2 | 0.5% |
| System Memory | Host | 2 | 0.5% |

*Table 2.* **Root-cause categorization of unexpected interruptions during a 54-day period of Llama-3 405B pre-training.** (Dubey et al., 2024) About 78% of unexpected interruptions were attributed to confirmed or suspected hardware issues.

*Table 3.* Type of GPU Xid Errors and Its Causes

| Xid Errors | Analysis |
|---|---|
| Software Causes: Xid_13/31 Xid_43/45 | Triggered by application programs, software-related Xid messages may indicate anomalies in GPU memory affecting code and data segments. However, it's crucial to consider other information for a comprehensive hardware functionality assessment. |
| NVLink Error: Xid 74 | Xid74 indicates errors in NVLink. For PCIe A100, it mainly occurs on the NVLink Bridge between two GPUs. Its occurrence rate is several orders of magnitude higher than other hardware faults. Apart from stress testing to exclude those that are constantly repeating errors, there isn't a good way to avoid the occurrence of Xid74 issues. |
| Memory ECC Error: Xid_63/64 Xid_94/95 | Triggered when the GPU handles memory ECC errors on the GPU. With the introduction of row remapping technology in A100, most instances can be resolved by simply resetting the GPU to retain optimal performance. |
| Uncorrectable GPU Failures: Xid_44/48 Xid_61/62/69/79 | These failures mean an uncorrectable error occurs on the GPU, which is also reported back to the user application. A GPU reset or node reboot is needed to clear this error. |
| Other Failures: Xid 119 | Xid119 means GPU GSP module failed. These failures need to undergo a field test, and most need to RMA. |

*Table 4.* **Raw Data** of GPU Xid Errors during one year in Fire-Flyer HPC (An et al., 2024)

| GPU Error Type | Xid Code | Number | Percentage |
|---|---|---|---|
| NVLink Error | xid_74 | 5521 | 42.57% |
| Software Causes | xid_13 | 45 | 0.35% |
| | xid_31 | 2487 | 19.18% |
| | xid_43 | 4342 | 33.48% |
| | xid_45 | 240 | 1.85% |
| GPU ECC Error | xid_63 | 245 | 1.89% |
| | xid_64 | 2 | 0.02% |
| | xid_94 | 13 | 0.10% |
| | xid_95 | 17 | 0.13% |
| Uncorrectable Failures | xid_44 | 1 | 0.01% |
| | xid_48 | 2 | 0.02% |
| | xid_61 | 13 | 0.10% |
| | xid_62 | 3 | 0.02% |
| | xid_69 | 1 | 0.01% |
| | xid_79 | 37 | 0.29% |
| GPU GSP ERROR | xid_119 | 1 | 0.01% |
| **Total** | | **12970** | **100.00%** |

## C. More Discussion

### C.1. Silent Data Corruption Errors

Silent data corruption (SDC) errors are particularly insidious in high-performance computing (HPC) (Wang et al., 2023a; He et al., 2023b), database (Bacon, 2022) and communication systems because they can go undetected and lead to incorrect results. These errors can occur due to various reasons, including hardware faults, software bugs, or cosmic radiation. In the context of HPC, SDC errors can significantly impact the reliability and accuracy of computations, especially in large-scale simulations and data-intensive applications. The large-scale distributed deep learning might be severely influenced by the SDC errors (He et al., 2023b).

In communication systems, SDC errors can be introduced during data transmission between nodes in a distributed computing environment (Fiala et al., 2012; łgorzata Steinder & Sethi, 2004). These errors can be caused by issues such as faulty network hardware, electromagnetic interference, or signal degradation over long distances. The impact of SDC errors in communication can be severe, as they can lead to incorrect data being propagated through the system, potentially causing widespread computational errors.

Table 2 shows the root-cause categorization of unexpected interruptions during a 54-day period of Llama 3 405B pre-training (Dubey et al., 2024). About 78% of unexpected interruptions were attributed to confirmed or suspected hardware issues, including faulty GPUs, GPU memory, and other components. These interruptions can lead to significant downtime and data loss, affecting the overall performance and reliability of the system. The SDC and network errors occupy a significant portion of the interruptions, highlighting the importance of addressing these issues in distributed computing environments. Note that the reported SDC erros belong to the explicit results that are obviously observed. However, there exists a large portion of silent errors with low error degree may appear in the training process, which is hard to detect and not reported.

There is a substantial amount of SDC in data center processors (He et al., 2023b; Wang et al., 2023a), leading to complex issues that are difficult to replicate and locate. In Fire-Flyer HPC (An et al., 2024), various computational errors and GPU memory errors not detected by Error Correction Code (ECC) listed in Table 3, which led to models' gradnorm spikes, loss explosions, and even nonconvergence. Tackling these silent errors is crucial for ensuring the reliability and accuracy of distributed training systems. The errors like Xid 63/64 will cause the failed convergence problems.

Table 4 shows that NVLink Erros and software errors occupy a large portion of all errors. It is crucial to address the SDC erros in both communication and computation.

### C.2. SDC Error Simulation

Fig. 11 shows the bias distribution with different noise degrees. For the $\sigma = 0.0001$, almost all elements are less than 3e-4. Fig. 12 shows the maximal value in the noise during training with different noise degrees. After each 500 iterations, there is a burst value happens, which is more significant for the larger noise degree.

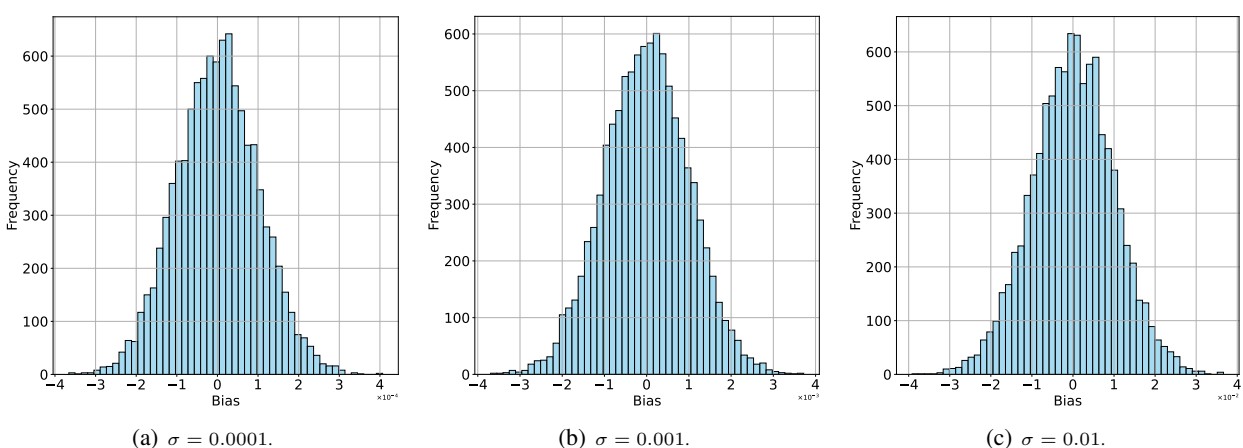

(a) $\sigma = 0.0001$.       (b) $\sigma = 0.001$.       (c) $\sigma = 0.01$.

*Figure 11.* The bias distribution for all elements in a gradient with different $\sigma^2$.

**Gradient Magnitude Distribution.** Fig. 13 and 14 show the distribution of values in the gradients of the ResNet-50 when

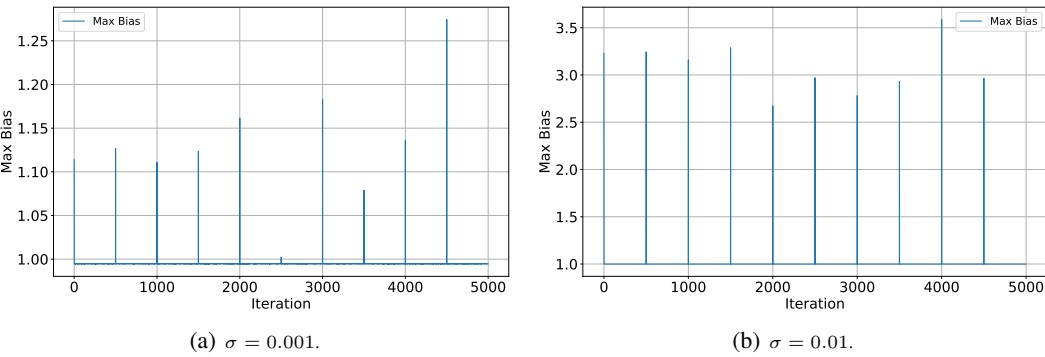

(a) $\sigma = 0.001$.  (b) $\sigma = 0.01$.

*Figure 12.* The maximal value in the noise during training.

training with CIFAR-100 at different iterations. Comparing the magnitudes of gradients with the noise, we can see that even the noise with $\sigma = 0.001$ is a large noise that has similar magnitude to the gradients. In real-world scenarios, noises with $\sigma \geq 0.01$ happen less. For the significantly larger noise error, which can be detected by some machine learning methods like the gradient clip, or the majority voting.

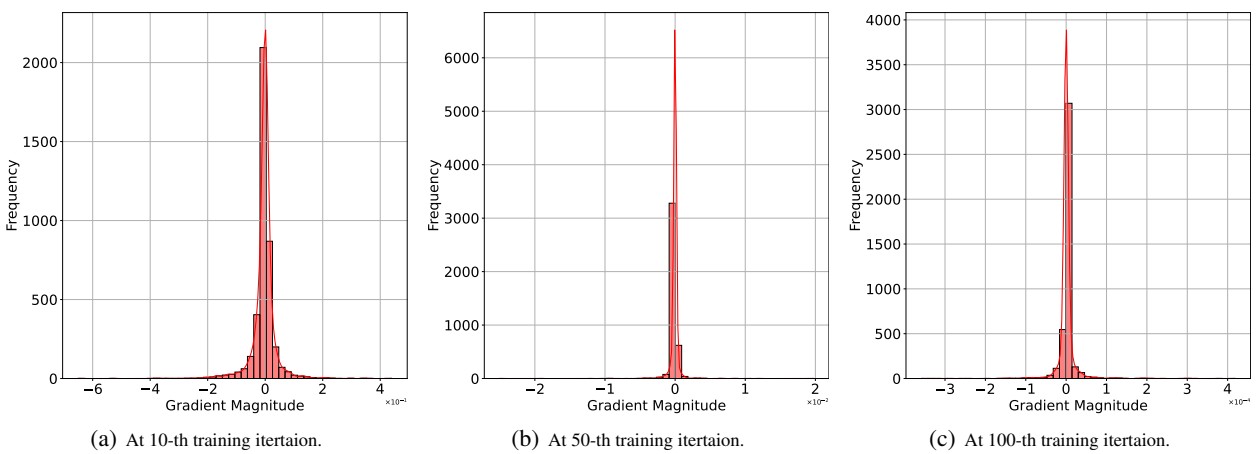

(a) At 10-th training itertaion.  (b) At 50-th training itertaion.  (c) At 100-th training itertaion.

*Figure 13.* The bias distribution for all elements of gradients of Conv layer in the first block.

## D. Proof

In this section, we provide the detailed proof of Lemma 3.1, Theorem 3.3, Lemma 4.2 and Theorem 4.3. We rewrite all of them in this section for the convenience of reading.

### D.1. Some Definitions and Assumptions

**Definition D.1.** *(Virtual Average). In distributed stochastic gradient descend (Eq. 6) with inconsistent gradient (Definition 2.1), an averaged model weight sequence $\{\bar{\theta}_t\}_{t \geq 0}$ is defined as*

$$\bar{\theta}_0 = \theta_0, \qquad \bar{\theta}_t = \frac{1}{M} \sum_{i=1}^{M} \theta_t^i. \tag{12}$$

From Definition 2.1, Eq. 6 and 12, we have

$$\bar{\theta}_{t+1} = \bar{\theta}_t - \eta_t \tilde{g}_t. \tag{13}$$

### D.2. Increasing Model Divergence

**Lemma D.2** (Increasing Model Divergence (Lemma 3.1)). *With the same initial point $\theta_0^m = \theta_0$ across workers $\{m | m = 1, 2, ..., M\}$, the DSGD with noise $\epsilon_t^m \sim \mathcal{N}(0, \sigma^2)$ introduces accumulated model divergence $\Delta_t^m$ along the*

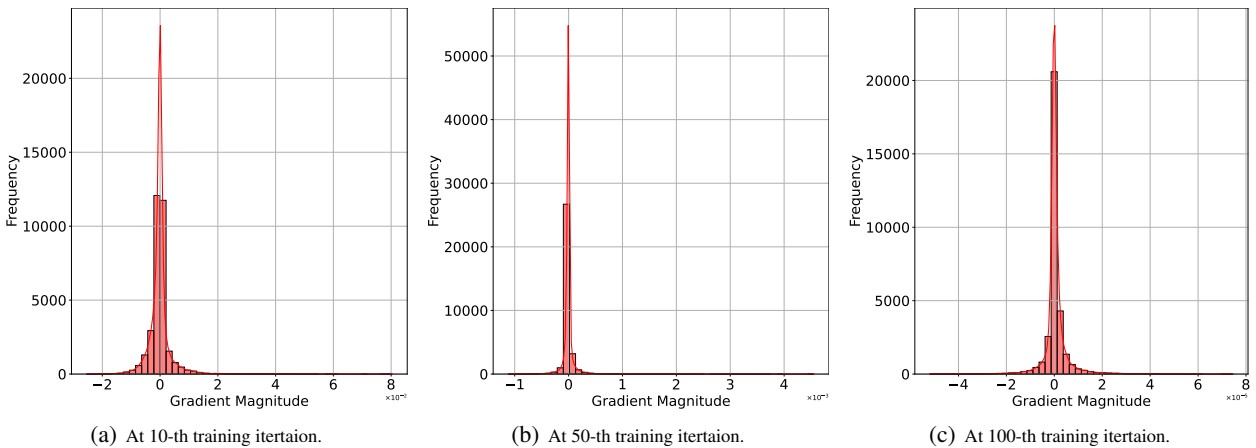

(a) At 10-th training itertaion.

(b) At 50-th training itertaion.

(c) At 100-th training itertaion.

*Figure 14.* The bias distribution for all elements of gradients of Conv layer in the second block.

*training process as*

$$\mathbb{E}||\bar{\theta}_{t+1} - \theta_{t+1}^m||^2 = \frac{(M+1)\sigma^2}{M} \sum_{s=0}^{t} \eta_s^2. \tag{14}$$

**Proof of Lemma 3.1.** We define the $\bar{\theta}_t = \frac{1}{M} \sum_{i=1}^{M} \theta_t^i$ and $\tilde{g}_t = \frac{1}{M} \sum_{i=1}^{M} \tilde{g}_t^i = \frac{1}{M} \sum_{i=1}^{M} (\bar{g}_t + \epsilon_t^m)$. Then, we have $\bar{\theta}_{t+1} = \bar{\theta}_t - \eta_t \tilde{g}_t$. By substituting Eq. 5 and 6 into $\Delta_t^m$ and iterating.

$$\begin{aligned}
\mathbb{E}||\bar{\theta}_{t+1} - \theta_{t+1}^m||^2 &= \mathbb{E}||\bar{\theta}_t - \eta_t \tilde{g}_t - \theta_t^m + \eta_t \tilde{g}_t^m||^2 \\
&= \mathbb{E}||\bar{\theta}_t - \theta_t^m||^2 + \eta_t^2 \mathbb{E}||\tilde{g}_t - \tilde{g}_t^m||^2 \\
&\quad + 2\eta_t \underbrace{\mathbb{E}\langle \bar{\theta}_t - \theta_t^m, \tilde{g}_t^m - \tilde{g}_t \rangle}_{=0}.
\end{aligned} \tag{15}$$

By iterating above equation from $t \to 0$, we have

$$\begin{aligned}
\mathbb{E}||\bar{\theta}_{t+1} - \theta_{t+1}^m||^2 &= \underbrace{\mathbb{E}||\bar{\theta}_0 - \theta_0^m||^2}_{=0} + \sum_{s=0}^{t} \eta_s^2 \mathbb{E}||\tilde{g}_s - \tilde{g}_s^m||^2 \\
&= \sum_{s=0}^{t} \eta_s^2 \mathrm{Var}(\frac{1}{M} \sum_{k=1}^{M} (\bar{g}_t + \epsilon_t^k) - (\bar{g}_t + \epsilon_t^m)) \\
&= \sum_{s=0}^{t} \eta_s^2 \mathrm{Var}(\frac{1}{M} \sum_{k=1}^{M} \epsilon_s^k - \epsilon_s^m) \\
&= \frac{(M+1)\sigma^2}{M} \sum_{s=0}^{t} \eta_s^2
\end{aligned}$$

$\square$

### D.3. Convergence with noised training.

Firstly, we provide the Lemma D.3 before proving Theorem 3.3.

**Lemma D.3.** *Let $\{\theta_t\}_{t \geq 0}$ and $\{\bar{\theta}_t\}_{t \geq 0}$ for $m \in [M]$ be defined as in Equation (8), (12) and let f be L-smooth and $\mu$-strongly*

*convex and $\eta_t \leq \frac{1}{4L}$. Then*

$$
\begin{aligned}
\mathbb{E}||\bar{\theta}_{t+1} - \theta^*||^2 \leq & (1 - \mu\eta_t)\mathbb{E}||\bar{\theta}_t - \theta^*||^2 + \eta_t^2 \mathbb{E}||\tilde{g}_t - \nabla F_t||^2 \\
& - \frac{1}{2}\eta_t \mathbb{E}(f(\bar{\theta}_t) - f^*) + \frac{2L\eta_t}{M}\sum_{i=1}^{M}\mathbb{E}||\bar{\theta}_t - \theta_t^i||^2
\end{aligned}
\tag{16}
$$

**Proof of Lemma D.3.** Using the update Equation 13, we have

$$
\begin{aligned}
||\bar{\theta}_{t+1} - \theta^*||^2 = & ||\bar{\theta}_t - \eta_t\tilde{g}_t - \theta^*||^2 \\
= & ||\bar{\theta}_t - \eta_t\tilde{g}_t - \theta^* - \eta_t\nabla F_t + \eta_t\nabla F_t||^2 \\
= & ||\bar{\theta}_t - \eta_t\nabla F_t - \theta^*||^2 + \eta_t^2||\tilde{g}_t - \nabla F_t||^2 \\
& + 2\eta_t\langle\bar{\theta}_t - \theta^* - \eta_t\nabla F_t, \tilde{g}_t - \nabla F_t\rangle.
\end{aligned}
\tag{17}
$$

Observe that

$$
\begin{aligned}
& ||\bar{\theta}_t - \eta_t\nabla F_t - \theta^*||^2 \\
= & ||\bar{\theta}_t - \theta^*||^2 + \eta_t^2||\nabla F_t||^2 - 2\langle\bar{\theta}_t - \theta^*, \eta_t\nabla F_t\rangle \\
\leq & ||\bar{\theta}_t - \theta^*||^2 + \eta_t^2\frac{1}{M}\sum_{i=1}^{M}||g(\theta_t^i)||^2 \\
& - \frac{2\eta_t}{M}\sum_{i=1}^{M}\langle\bar{\theta}_t - \theta_t^i + \theta_t^i - \theta^*, g(\theta_t^i)\rangle \\
= & ||\bar{\theta}_t - \theta^*||^2 + \eta_t^2\frac{1}{M}\sum_{i=1}^{M}||g(\theta_t^i) - g(\theta^*)||^2 \\
& - \frac{2\eta_t}{M}\sum_{i=1}^{M}\langle\theta_t^i - \theta^*, g(\theta_t^i)\rangle - \frac{2\eta_t}{M}\sum_{i=1}^{M}\langle\bar{\theta}_t - \theta_t^i, g(\theta_t^i)\rangle.
\end{aligned}
\tag{18}
$$

By $L$-smoothness, we have

$$
||g(\theta_t^i) - g(\theta^*)||^2 \leq 2L(f(\theta_t^i) - f^*).
\tag{19}
$$

By $\mu$-strong convexity, we have

$$
-\langle\theta_t^i - \theta^*, g(\theta_t^i)\rangle \leq -(f(\theta_t^i) - f^*) - \frac{\mu}{2}||\theta_t^i - \theta^*||^2.
\tag{20}
$$

To estimate the last term in (18), we use $2\langle a, b\rangle \leq \gamma||a||^2 + \gamma^{-1}||b||^2$ for $\gamma > 0$, thus

$$
\begin{aligned}
-2\langle\bar{\theta}_t - \theta_t^i, g(\theta_t^i)\rangle \leq & 2L||\bar{\theta}_t - \theta_t^i||^2 + \frac{1}{2L}||g(\theta_t^i)||^2 \\
= & 2L||\bar{\theta}_t - \theta_t^i||^2 + \frac{1}{2L}||g(\theta_t^i) - g(\theta^*)||^2 \\
\leq & 2L||\bar{\theta}_t - \theta_t^i||^2 + (f(\theta_t^i) - f^*).
\end{aligned}
\tag{21}
$$

By applying these estimates to (18), we get

$$
\begin{aligned}
& ||\bar{\theta}_t - \theta^* - \eta_t\nabla F_t||^2 \\
\leq & ||\bar{\theta}_t - \theta^*||^2 + \frac{2\eta_t L}{M}\sum_{i=1}^{M}||\bar{\theta}_t - \theta_t^i||^2 \\
& + \frac{2\eta_t}{M}\sum_{i=1}^{M}((\eta_t L - \frac{1}{2})(f(\theta_t^i) - f^*) - \frac{\mu}{2}||\theta_t^i - \theta^*||^2)
\end{aligned}
\tag{22}
$$

For $\eta_t \leq \frac{1}{4L}$ it holds $(\eta_t L - \frac{1}{2}) \leq -\frac{1}{4}$. By convexity of $a(f(\theta) - f^*) + b||\theta - \theta^*||^2$ for $a, b \geq 0$,

$$
\begin{aligned}
-\frac{1}{M} \sum_{i=1}^{M} (a(f(\theta_t^i) - f^*) + b||\theta_t^i - \theta^*||^2) \\
\leq -(a(f(\bar{\theta}_t) - f^*) + b||\bar{\theta}_t - \theta^*||^2).
\end{aligned}
\tag{23}
$$

Hence, we can continue in (22) and obtain

$$
\begin{aligned}
||\bar{\theta}_t - \theta^* - \eta_t \nabla F_t||^2 \leq (1 - \mu\eta_t)||\bar{\theta}_t - \theta^*||^2 \\
- \frac{1}{2}\eta_t (f(\bar{\theta}_t) - f^*) + \frac{2\eta_t L}{M} \sum_{i=1}^{M} ||\bar{\theta}_t - \theta_t^i||^2
\end{aligned}
\tag{24}
$$

Finally, we can combine (24) with (17). By taking expectation, we get

$$
\begin{aligned}
\mathbb{E}||\bar{\theta}_{t+1} - \theta^*||^2 \leq (1 - \mu\eta_t)\mathbb{E}||\bar{\theta}_t - \theta^*||^2 + \eta_t^2 \mathbb{E}||\tilde{g}_t - \nabla F_t||^2 \\
- \frac{1}{2}\eta_t \mathbb{E}(f(\bar{\theta}_t) - f^*) + \frac{2L\eta_t}{M} \sum_{i=1}^{M} \mathbb{E}||\bar{\theta}_t - \theta_t^i||^2
\end{aligned}
\tag{25}
$$

$\square$

Now, we can prove Theorem 3.3 with the help of Lemma D.3.

**Theorem D.4** (Convergence with noised training (Theorem 3.3.). *With object function defined in Eq. 1 satisfying Assumption 3.2, DSGD with noise $\epsilon_t^m \sim \mathcal{N}(0, \sigma^2)$ has the following convergence bound*

$$
\begin{aligned}
\frac{1}{T} \sum_{t=0}^{T-1} \eta_t \mathbb{E}(f(\bar{\theta}_t) - f^*) \leq &\frac{2\mathbb{E}||\bar{\theta}_0 - \theta^*||^2}{T} + \frac{2(\sigma_g^2 + \sigma^2)}{TM} \sum_{t=0}^{T-1} \eta_t^2 \\
&+ \frac{4L\sigma^2(M+1)}{TM} \sum_{t=0}^{T-1} \eta_t \sum_{s=0}^{t-1} \eta_s^2.
\end{aligned}
$$

### D.4. Bounded Model Divergence

**Lemma D.5** (Bounded Model Divergence (Lemma 4.2). *If $gap(\mathcal{A}) \leq H$ and sequence of decreasing positive stepsizes $\{\eta_t\}_{t \geq 0}$ satisfying $\eta_t \leq 2\eta_{t+H}$ for all $t \geq 0$, then. With the same initial point $\theta_0^m = \bar{\theta}_0$ across workers $\{m|m = 1, 2, ..., M\}$, the DSGD with noise $\epsilon_t^m \sim \mathcal{N}(0, \sigma^2)$ introduces accumulated model divergence $\Delta_t^m$ along the training process as*

$$
\mathbb{E}||\bar{\theta}_{t+1} - \theta_{t+1}^m||^2 \leq \frac{4H(M+1)\sigma^2\eta_t^2}{M}
\tag{26}
$$

**Proof of Lemma 4.2.** By Lemma 3.1, and observing that all $\theta_{t+1}^m$ will be synchronized at the synchronization point as Eq. 8 or Eq. 11, we have

$$
\mathbb{E}||\bar{\theta}_r - \theta_r^m||^2 = 0,
$$

where $r = H_t \leq \lfloor t/H \rfloor$ represents the last synchronization timestamp until iteration $t$. Thus, we have the following equation by iterating Eq. 15 from $t \to r$,

$$\mathbb{E}||\bar{\theta}_{t+1} - \theta_{t+1}^m||^2 = \underbrace{\mathbb{E}||\bar{\theta}_r - \theta_r^m||^2}_{=0} + \sum_{s=r}^{t} \eta_s^2 \mathbb{E}||\tilde{g}_s - \tilde{g}_s^m||^2$$

$$= \sum_{s=r}^{t} \eta_s^2 \text{Var}(\frac{1}{M}\sum_{k=1}^{M}\epsilon_s^k - \epsilon_s^m)$$

$$= \frac{(M+1)\sigma^2}{M} \sum_{s=r}^{t} \eta_s^2 \mathbb{E}||\bar{\theta}_{t+1} - \theta_{t+1}^m||^2$$

$$= \frac{(M+1)\sigma^2}{M} \sum_{s=r}^{t} \eta_s^2 \leq \frac{4H(M+1)\sigma^2\eta_t^2}{M},$$

We use $\eta_t \leq \eta_r$ for $t \geq r$ and learning rate decay assumption $\eta_r \leq 2\eta_{r+H}$. Note that different learning rate schedule methods do not influence the order of this bound too much. $\qquad\square$

**Proof of Theorem 3.3.** By Equation (25), when $\mu = 0$, and $f$ is convex, we have

$$\mathbb{E}||\bar{\theta}_{t+1} - \theta^*||^2 \leq \mathbb{E}||\bar{\theta}_t - \theta^*||^2 + \eta_t^2 \mathbb{E}||\tilde{g}_t - \nabla F_t||^2$$
$$- \frac{1}{2}\eta_t\mathbb{E}(f(\bar{\theta}_t) - f^*) + \frac{2L\eta_t}{M}\sum_{i=1}^{M}\mathbb{E}||\bar{\theta}_t - \theta_t^i||^2 \qquad (27)$$

Rearranging Eq. 27, we have

$$\eta_t\mathbb{E}(f(\bar{\theta}_t) - f^*) \leq 2(\mathbb{E}||\bar{\theta}_t - \theta^*||^2 - \mathbb{E}||\bar{\theta}_{t+1} - \theta^*||^2)$$
$$+ 2\eta_t^2\mathbb{E}||\tilde{g}_t - \nabla F_t||^2 + \frac{4\eta_t L}{M}\sum_{i=1}^{M}\mathbb{E}||\bar{\theta}_t - \theta_t^i||^2 \qquad (28)$$

By summing $t$ from 0 to $T-1$,

$$\frac{1}{T}\sum_{t=0}^{T-1}\eta_t\mathbb{E}(f(\bar{\theta}_t) - f^*) \leq \frac{2\mathbb{E}||\bar{\theta}_0 - \theta^*||^2}{T} + \frac{2}{T}\sum_{t=0}^{T-1}\eta_t^2\mathbb{E}||\tilde{g}_t - \nabla F_t||^2$$
$$+ \frac{4L}{MT}\sum_{t=0}^{T-1}\eta_t\sum_{i=1}^{M}\mathbb{E}||\bar{\theta}_t - \theta_t^i||^2. \qquad (29)$$

For gradient estimation error from the noise, we have

$$\mathbb{E}||\tilde{g}_t - \nabla F_t||^2 = \mathbb{E}||\frac{1}{M}\sum_{i=1}^{M}g_i(\theta_t^i) + \frac{1}{M}\sum_{i=1}^{M}\epsilon_t^i - \frac{1}{M}\sum_{i=1}^{M}g(\theta_t^i)||^2$$

$$= \mathbb{E}||\frac{1}{M}\sum_{i=1}^{M}g_i(\theta_t^i) - \frac{1}{M}\sum_{i=1}^{M}g(\theta_t^i)||^2 + \mathbb{E}||\frac{1}{M}\sum_{i=1}^{M}\epsilon_t^i||^2$$

$$+ \underbrace{\frac{1}{M}\sum_{i=1}^{M}\mathbb{E}\langle g_i(\theta_t^i) - g(\theta_t^i), \epsilon_t^i\rangle}_{=0} \qquad (30)$$

$$= \frac{1}{M^2}\sum_{i=1}^{M}\mathbb{E}||g_i(\theta_t^i) - g(\theta_t^i)||^2 + \frac{1}{M^2}\sum_{i=1}^{M}\mathbb{E}||\epsilon_t^i||^2$$

$$\leq \frac{\sigma_g^2 + \sigma^2}{M}$$

Combining Eq. 30 and Lemma 4.2 into Eq. 29, we have

$$
\begin{aligned}
\frac{1}{T} \sum_{t=0}^{T-1} \eta_t \mathbb{E}(f(\bar{\theta}_t) - f^*) \leq & \frac{2\mathbb{E}||\bar{\theta}_0 - \theta^*||^2}{T} + \frac{2(\sigma_g^2 + \sigma^2)}{TM} \sum_{t=0}^{T-1} \eta_t^2 \\
& + \frac{4L\sigma^2(M+1)}{TM} \sum_{t=0}^{T-1} \eta_t \sum_{s=0}^{t-1} \eta_s^2,
\end{aligned}
\tag{31}
$$

which completes the proof. □

## D.5. Convergence with noised training with `PAFT-Sync`.

Here, we use the Martingale Lemma (Lemma 3.3 in (Stich et al., 2018)) to help our proof.

**Lemma D.6.** *Let $\{a_t\}_{t\geq 0}, a_t \geq 0, \{e_t\}_{t\geq 0}, e_t \geq 0$ be sequences satisfying*

$$
a_{t+1} \leq (1 - \mu\eta_t)a_t - \eta_t e_t A + \eta_t^2 B + \eta_t^3 C,
\tag{32}
$$

*for $\eta_t = \frac{4}{\mu(a+t)}$ and constants $A > 0, B, C \geq 0, \mu > 0, a > 1$. Then we have*

$$
\frac{A}{S_T} \sum_{t=1}^{T-1} w_t e_t \leq \frac{\mu a^3}{4S_T} a_0 + \frac{2T(T + 2a)}{\mu S_T} B + \frac{16T}{\mu^2 S_T} C,
\tag{33}
$$

*for $w_t = (a + t)^2$ and $S_T \triangleq \sum_{t=0}^{T-1} w_t = \frac{T}{6}(2T^2 + 6aT - 3T + 6a^2 - 6a + 1) \geq \frac{1}{3}T^3$.*

**Theorem D.7** (Convergence with noised training with `PAFT-Sync` ( 4.3).)**.** *With object function defined in Eq. 1 satisfying Assumption 3.2, DSGD with `PAFT` (Eq. 8 or 11) noise $\epsilon_t^m \sim \mathcal{N}(0, \sigma^2)$, we have,*

$$
\begin{aligned}
\mathbb{E}f(\hat{\theta}_T) - f^* \leq & \frac{\mu a^3}{2S_T}||\theta_0 - \theta^*||^2 + \frac{4T(T + 2a)(\sigma_g^2 + \sigma^2)}{\mu M S_T} \\
& + \frac{256T}{\mu^2 S_T} \frac{(M+1)}{M} \sigma^2 HL
\end{aligned}
$$

*where $\hat{\theta}_T = \frac{1}{MS_T} \sum_{m=1}^{M} \sum_{t=0}^{T-1} w_t \theta_t^m$, for $w_t = (a + t)^2$ and $S_T = \sum_{t=0}^{T-1} w_t \geq \frac{1}{3}T^3$*

**Proof of Theorem 4.3.** Using Lemma D.3, Eq. 30, Lemma 4.2 we get

$$
\begin{aligned}
\mathbb{E}||\bar{\theta}_{t+1} - \theta^*||^2 \leq & (1 - \mu\eta_t)\mathbb{E}||\bar{\theta}_t - \theta^*||^2 + \frac{\sigma_g^2 + \sigma^2}{M}\eta_t^2 \\
& - \frac{1}{2}\eta_t \mathbb{E}(f(\bar{\theta}_t) - f^*) + \frac{8LH\sigma^2(M+1)}{M}\eta_t^3
\end{aligned}
\tag{34}
$$

By Lemma D.6 and the convexity of $f$, rearranging Eq. 34, we have

$$
\begin{aligned}
\mathbb{E}f(\hat{\theta}_T) - f^* \leq & \frac{\mu a^3}{2S_T}||\theta_0 - \theta^*||^2 + \frac{4T(T + 2a)(\sigma_g^2 + \sigma^2)}{\mu M S_T} \\
& + \frac{256T}{\mu^2 S_T} \frac{(M+1)}{M} \sigma^2 HL
\end{aligned}
\tag{35}
$$

□

# E. More Experimental Results

## E.1. Eliminate Model Divergence

Fig. 15 shows that the in noised DSGD, the model divergence is accumulated during trainig, thus severely influencing convergence. While the PAFT can effectively illuminate the model divergence periodically, thus improving the convergence.

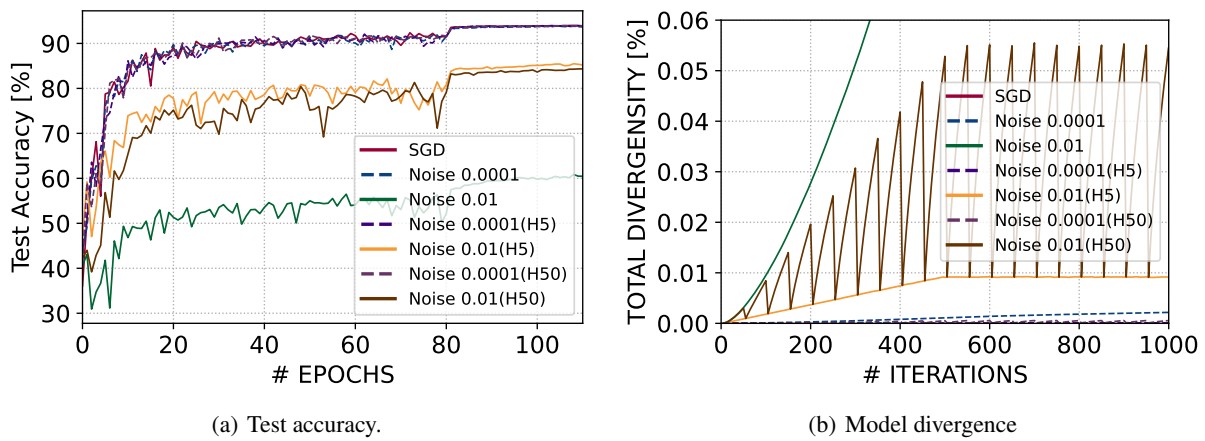

(a) Test accuracy.

(b) Model divergence

*Figure 15.* Training ResNet-18 with 4 workers.

## E.2. Convergence under Larger Noise

Fig. 16 and 17 show results of training ResNet-18, ResNet-50 and LLMs with larger noise degrees ($\sigma^2 = 0.1$ or $1.0$). Under the more severe noises, the convergence of LLMs is significantly influenced. And it is more difficult for PAFT to mitigate these erros. Nevertheless, such a large noise degree is not common in practice.

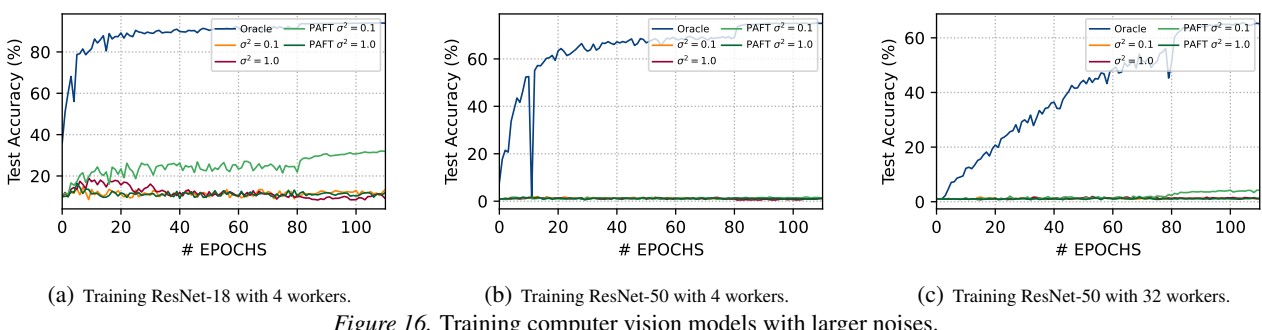

(a) Training ResNet-18 with 4 workers.     (b) Training ResNet-50 with 4 workers.     (c) Training ResNet-50 with 32 workers.

*Figure 16.* Training computer vision models with larger noises.

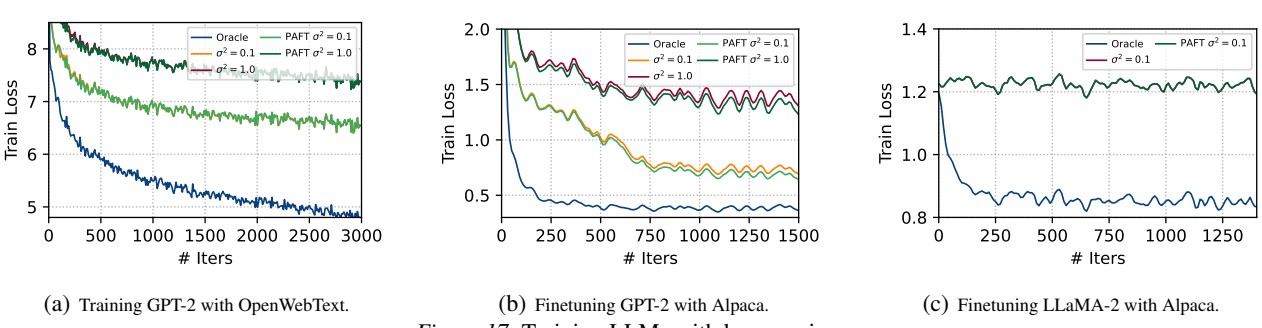

(a) Training GPT-2 with OpenWebText.     (b) Finetuning GPT-2 with Alpaca.     (c) Finetuning LLaMA-2 with Alpaca.

*Figure 17.* Training LLMs with larger noises.

### E.3. Comparing Synchronizing Optimizer States

Fig. 18 provides results of comparing `PAFT` with synchronizing model or all parameters (including optimizer states). The results show that synchronizing all parameters can improve the convergence than synchronizing model only. However, the improvement is limited, and the overhead of synchronizing all parameters is much higher than synchronizing model only. Thus, synchronizing model only is more practical in distributed training.

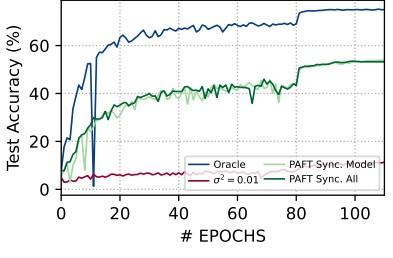
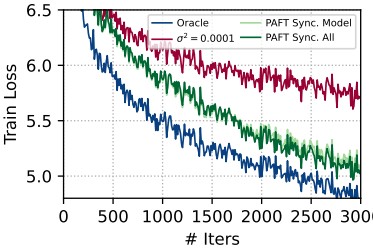
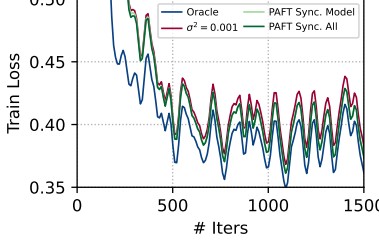

(a) Training ResNet-50.  (b) Training GPT-2 with OpenWebText.  (c) Training GPT-2 with Alpaca.

*Figure 18.* Comparing `PAFT` with synchronizing the model or all parameters (including optimizer states). The "Sync. All" denotes synchronizing all parameters, including optimizer states.

