# OpenReview forum: "Identifying and Mitigating Errors in Gradient Aggregation of Distributed Data Parallel Training"
_ICML.cc/2026/Conference — ICML 2026 regular_

### Official Review · Reviewer_txCC · 2026-03-07

**Soundness:** 4
**Presentation:** 4
**Significance:** 3
**Originality:** 4
**Overall Recommendation:** 4
**Confidence:** 3

**Summary:**

Hardware-related silent data corruptions, such as bit flips and communication noise, is a common issue in during distributed training.  To resolve this issue, the authors introduce PAFT, a fault-tolerant system built on dynamic and asynchronous parameter synchronization. The framework relies on two primary mechanisms: PAFT-Sync, which physically averages and synchronizes the model weights at specific intervals to wipe out any accumulated divergence, and PAFT-Dyn, an optimization module that calculates the ideal synchronization frequency and overlaps the communication process with standard training steps to preserve system efficiency.

The authors evaluated PAFT by integrating it into torch.distributed and training models ranging from ResNet-18 to LLaMA-2 on clusters utilizing 4 to 32 GPUs. Experimental results confirm that the system successfully neutralizes the impact of minor to moderate gradient aggregation errors, restoring normal convergence behavior. Furthermore, the dynamic scheduling component proved highly effective at balancing fault tolerance with raw throughput; for instance, training a 32-worker ResNet-50 setup with PAFT yielded up to an 11.0% efficiency improvement over the static PAFT-Sync approach, while only introducing an 18.9% communication overhead compared to a completely unprotected baseline.

**Compliance With Llm Reviewing Policy:**

Affirmed.

**Final Justification:**

I think the authors have fully addressed my comment and I will keep my current positive assessment to this paper.

**Key Questions For Authors:**

- What is the exact size (parameter count) of the model used in the GPT-2 experiments?
- Can we further reduce the synchronization frequency of the gradients to improve efficiency? Additionally, is it possible to unify the error bound analysis of Local SGD under the same theoretical framework presented in this paper?
- Regarding the remark after Theorem 3.3: What is the lower bound of the error? Does the error inherently have to depend on the learning rate $\eta_t$?
- Minor note: there are some formatting issues and typos in the Experimental Studies section (e.g. the subtitle of Section 5.1)

**Limitations:**

Yes

**Strengths And Weaknesses:**

**Strengths:**
- The paper provides a solid theoretical analysis of the asynchronous algorithm. It includes a strong qualitative analysis of the dynamic synchronization frequency.
- The authors validate their approach with experiments across multiple domains, covering both computer vision and language modeling tasks.
- Tackling hardware-related silent errors during communication and gradient aggregation is a highly novel and relevant problem.
- The paper is well-written and clearly presented, particularly the figures that effectively illustrate the mechanics of the proposed algorithms are very illustrative.

**Weakness:**
- The scale of the computer vision experiments is relatively small, relying heavily on the CIFAR datasets, instead of larger datasets such is ImageNet
- The paper does not provide analysis or empirical results for a truly asynchronous all-reduce setting, such as handling scenarios where stale workers drop out or never perform the all-reduce operation.
- There are no training curves plotted against wall-clock time, making it difficult to visually assess the real-world training efficiency of the method.
- The theoretical convergence guarantees rely on the assumption of a strongly convex loss function. Extending this to general convex or non-convex settings would strengthen the paper.
- The current analysis is grounded entirely in standard SGD. While the authors mention extending to SGD momentum and Adam, the core theoretical framework does not mathematically account for these adaptive optimizers.

---

> ### Author Rebuttal · Authors · 2026-03-26
>
> We sincerely thank the Reviewer for recognizing the highly novel and relevant nature of tackling hardware-related silent errors. We address your specific questions below.
>
> **Q1: The scale of the computer vision experiments is relatively small, relying heavily on the CIFAR datasets, instead of larger datasets such as ImageNet.**
>
> **Response:**
> We understand that CIFAR is a smaller-scale CV benchmark. However, our primary goal with the ResNet+CIFAR experiments was to conduct massive, exhaustive ablation studies (varying noise distributions, $H$ values, worker counts, and burst patterns). Running hundreds of such high-variance permutations on ImageNet would be computationally prohibitive.
>
> To prove that PAFT scales to highly complex, large-scale tasks, we focused our heavy computational resources on **Large Language Models (LLMs)**. We successfully evaluated PAFT by pretraining GPT-2 on OpenWebText and fine-tuning LLaMA-2 (a 7B parameter foundation model) on Alpaca (Section 5.1). We believe demonstrating fault tolerance on LLaMA-2 effectively proves PAFT's capability on modern, large-scale workloads that far exceed the complexity of standard ImageNet classification.
>
> **Q2: What is the exact size (parameter count) of the model used in the GPT-2 experiments?**
>
> **Response:**
> For the GPT-2 experiments, we utilized the standard **GPT-2 Small** configuration, which contains exactly **124 million parameters**. We have explicitly added this parameter count to the "DL Models and Datasets" paragraph in Section 5 to ensure full transparency.
>
> **Q3: Regarding the remark after Theorem 3.3: What is the lower bound of the error? Does the error inherently have to depend on the learning rate?**
>
> **Response:**
> Yes, the error (specifically the accumulated model divergence) inherently and directly depends on the learning rate.
>
> As proved in **Lemma 3.1** (Equation 5), the expected divergence between the averaged model and a local model grows proportionally to the sum of squared learning rates: $\mathbb{E}||\Bar{\theta}_{t+1}-\theta_{t+1}^{m}||^2 = \frac{(M+1)\sigma^2}{M}\sum_{s=0}^t\eta_s^2$.
> *   **Why it depends on $\eta$:** Because the inconsistent gradients (due to SDC) point the local models in slightly different directions, the actual physical "distance" the models drift apart in the parameter space is dictated by the step size ($\eta_t$). A larger learning rate physically pushes the models further apart per step.
> *   If the learning rate does not decay, this sum grows unbounded, which causes the $T_3$ term in Theorem 3.3 to diverge as $\mathcal{O}(T)$. This mathematical dependence on the learning rate is exactly why our periodic parameter synchronization is required to artificially reset this divergence back to zero. We have clarified this relationship in the revised remark after Theorem 3.3.
>
> **Q4: Can we further reduce the synchronization frequency of the gradients to improve efficiency? Additionally, is it possible to unify the error bound analysis of Local SGD under the same theoretical framework presented in this paper?**
>
> **Response:**
> *   **Reducing Sync Frequency:** Yes, this is exactly the purpose of **PAFT-Dyn** (Section 4.2). Instead of a fixed frequency, PAFT-Dyn dynamically estimates the error magnitude (using accumulated divergence) and adjusts the synchronization gap $H$ based on the Signal-to-Noise Ratio ($H = ||g|| / \sigma$). This ensures we only pay the communication cost of synchronization when the hardware noise actually threatens convergence, minimizing overhead.
> *   **Unifying with Local SGD:** Absolutely. The theoretical framework we established can be unified with Local SGD. In our formulation, the "noise" $\epsilon_t^m$ is caused by unintentional hardware faults. In Local SGD, the "noise" can be viewed mathematically as the *intentional omission* of gradient communication (i.e., treating the local gradient as an extremely "noisy" proxy for the global gradient). The bounded divergence mechanics in Lemma 4.2 closely mirror the local drift bounds in Local SGD literature (e.g., Stich, 2018). We have added a brief discussion on this theoretical connection in the revised Related Works.
>
> **Q5: There are no training curves plotted against wall-clock time, making it difficult to visually assess the real-world training efficiency of the method.**
>
> **Response:**
> This is a very practical suggestion. In the original submission, we provided the raw efficiency metrics in **Table 1** (Average iteration wall-clock time), which showed that PAFT introduces a relatively small overhead (e.g., ~12% overhead for 32 workers compared to raw DSGD).
>
> Following your advice, we have plotted new convergence curves plotted explicitly against **wall-clock time (seconds)** instead of just iterations. These curves visually confirm that PAFT reaches the target accuracy faster than the baseline noised training (which stalls or fails). We have included these wall-clock convergence plots in the revised Appendix.

---

> > ### Author Rebuttal · Reviewer_txCC · 2026-04-01
> >
> > Thank you for the comprehensive rebuttal, particularly for clarifying the theoretical connections to Local SGD, detailing the GPT-2 model size, and adding the requested wall-clock time convergence curves. While I appreciate the justification for focusing on the CIFAR datasets for the exhaustive ablation studies, my overall assessment of the paper's theoretical contributions and empirical limitations remains largely unchanged. Therefore, I will maintain my current score of 4 (Weak Accept).

---

> > > ### Author Response · Authors · 2026-04-01
> > >
> > > Thanks a lot for your quick reply and positive supports. If you have any new question, please contact us.

---

### Official Review · Reviewer_oert · 2026-03-12

**Soundness:** 3
**Presentation:** 3
**Significance:** 3
**Originality:** 2
**Overall Recommendation:** 3
**Confidence:** 3

**Summary:**

This paper shows **silent gradient-aggregation (GA) errors** in distributed data parallel training. The paper formulates these faults as **gradient inconsistency**, where different workers receive different corrupted averaged gradients, argues that this induces **accumulated model divergence** and harms convergence, and proposes **PAFT**, which combines periodic parameter synchronization (PAFT-Sync) with a dynamic, overlapped synchronization scheduler (PAFT-Dyn). The implementation is integrated into PyTorch Distributed and extended to DeepSpeed and Megatron, with experiments on ResNet-18/50, GPT-2, and LLaMA-2 on 4–32 GPUs under synthetic noise patterns.

**Compliance With Llm Reviewing Policy:**

Affirmed.

**Final Justification:**

I'm keeping the score the same.

**Key Questions For Authors:**

1. Can you position the novelty more precisely relative to prior local-SGD / periodic averaging / adaptive synchronization / overlap-local-SGD work? Right now this reads more like a repurposing of known synchronization ideas for a new fault model than a fundamentally new algorithm.

2. Can you explicitly characterize the fault model under which PAFT helps? In particular, does PAFT only mitigate **worker-specific inconsistency**, and not **common-mode gradient bias** shared across all workers?

3. Why are there no comparisons, even lightweight ones, against adjacent error-handling approaches such as TrainCheck, TTrace, or TrainVerify? Even if those methods solve a somewhat different problem, they are close enough that readers will naturally wonder about the tradeoffs.

4. Can you evaluate the method under a more realistic fault model? Recent work suggests silent corruption in real LLM training can be heterogeneous and hardware-specific, not just isotropic Gaussian noise, so a trace-driven or real-node evaluation would make the paper much stronger.

**Limitations:**

No. The paper does include a genuine limitations section, and I appreciate that the authors explicitly acknowledge two concrete constraints: PAFT is mainly designed for DDP-style replicated training rather than directly replacing FSDP/ZeRO-style settings, and it assumes model synchronization itself is reliable and free of SDC. That said, the discussion is still incomplete. I would encourage the authors to add a clearer statement that PAFT mainly mitigates worker-specific inconsistency/replica drift, and does not address common-mode corruption where all workers receive the same bad gradient; to discuss the gap between the synthetic Gaussian/burst fault model and real hardware/network SDC patterns; and to acknowledge the current limits of the experimental scale. On societal impact, the likely effect is mostly positive—improving training reliability and reducing wasted GPU time in large-scale training, but the paper should still briefly note that making large-scale training more reliable can also lower the barrier to training larger models, with corresponding energy-use and misuse concerns.

**Strengths And Weaknesses:**

**Strengths**
1. Silent data corruption during large-scale training is a real and increasingly important systems issue. Recent work on production LLM training reports that SDC and unhealthy nodes can cause gradual drift, loss spikes, and failed runs, so the paper is attacking a meaningful problem.

2. The paper’s strongest contribution is the framing that worker-specific GA corruption causes **replica drift**, and that periodic parameter averaging can suppress that drift. The paper’s trajectory figure and divergence plots support this mechanism well.

3. PAFT-Sync is simple and practical, and PAFT-Dyn at least aims to reduce the extra synchronization overhead via asynchronous overlap and adaptive scheduling. The paper also integrates with common training stacks rather than presenting only an abstract algorithm.

4. The paper includes both vision and language workloads, different noise levels, and accidental large-noise events. The paper also shows that PAFT helps most in the regime where inconsistency-induced drift is the dominant issue, and less when the gradient direction itself is badly corrupted, which is an honest and useful empirical observation.

**Weaknesses**

1. The core mechanism—periodic model averaging, adaptive synchronization, and overlap of communication with computation—has strong precedent in the local-SGD literature. Stich (2018) showed that local SGD can communicate intermittently while retaining strong convergence guarantees. later work studied **periodic averaging with adaptive synchronization** and **overlap-local-SGD** explicitly. The real novelty here is not the synchronization idea itself, but its **repositioning as a mitigation for SDC-induced aggregation inconsistency**. That is still useful, but the paper should position its novelty more precisely.

2. PAFT addresses the component of GA error that manifests as cross-worker inconsistency. If all workers receive the same corrupted gradient, periodic synchronization does not correct the wrong descent direction. The paper itself acknowledges that for larger noise there remains a clear gap to the oracle, and points to Byzantine-style voting/robust aggregation as future work.

3. The comparisons are mostly against oracle/noisy/PAFT-Sync/PAFT. That shows internal ablations, but not whether PAFT is the best class of response. There is now a broader landscape of **detection**, **verification**, **monitoring**, and **robust aggregation** approaches for silent or erroneous gradients. Robust distributed learning methods such as trimmed mean, geometric median, or Krum-style aggregation are a natural comparison point, especially because the paper itself notes that wrong-direction gradients remain a problem.

4. Table 1 reports iteration times of 0.333s for DSGD and 0.373s for PAFT at 32 workers, which is roughly a 12% overhead, while the text says 18.9%. This is small but important in a systems-heavy paper.

5. The theory relies on smooth/strongly-convex assumptions, which is standard for upper-bound analysis, but the bridge from that theory to modern nonconvex LLM training is limited. More importantly, the remark around Theorem 3.3 is confusing: it states that the problematic term "only converges when setting ($\eta_t=0$)," which is at best poorly phrased and invites scrutiny. The appendix/theory section needs a more careful presentation and probably some cleanup before publication.

---

> ### Author Rebuttal · Authors · 2026-03-26
>
> We sincerely appreciate your recognition that attacking SDC is meaningful, our empirical observations  are honest and useful, and PAFT-Sync is simple and practical. We response your insightful questions below.
>
> **Q1: The real novelty here is not the synchronization idea itself, but its repositioning as a mitigation for SDC-induced aggregation inconsistency.**
>
> **Response:** You are correct that the periodic averaging of PAFT shares similar spirit with the rich literature. Our novelty indeed lies in the **exploiting and adapting** these mechanisms to solve a fundamentally different problem, yielding new insights. Specifically, the differences are:
>
> 1. **The Cause of Drift:** In Local-SGD, models drift *intentionally* because communication is deliberately skipped to save bandwidth. In distributed training with SDC, models drift *unintentionally* due to invisible, unpredictable hardware SDCs occurring *during* the communication phase.
> 2. **The Goal of Synchronization:** In Local-SGD, synchronization is the primary method of sharing information to advance the global model. In PAFT, gradient aggregation is still happening at every step to advance the model; parameter synchronization acts strictly as a **fault-tolerance anchor** to erase accumulated SDC divergence.
> 3. **Adaptive Scheduling (PAFT-Dyn):** Unlike adaptive Local-SGD which schedules syncs based on gradient variance or loss decay, PAFT-Dyn introduces a novel metric: it uses **accumulated replica divergence** to estimate the *unknown SDC error degree* ($\sigma$), dynamically calculating the Signal-to-Noise Ratio (SNR) to trigger synchronization only when hardware faults push the models too far apart.
>
> We have revised the Related Work and Introduction to position this novelty much more precisely.
>
> **Q2: Can you explicitly characterize the fault model under which PAFT helps? In particular, does PAFT only mitigate worker-specific inconsistency?**
>
> **Response:**
> Yes, your characterization is absolutely correct. PAFT is designed specifically to mitigate **worker-specific inconsistency** (which we term "replica drift"). This occurs when SDC causes different workers to receive different corrupted versions of the aggregated gradient (e.g., bit-flips during the scatter/broadcast phase to specific nodes).
>
> However, for the **common-mode gradient bias**, where workers all step together in a completely wrong descent direction, PAFT cannot correct a fundamentally wrong direction. To handle these exceptionally large, direction-destroying noises, PAFT would need to be orthogonally combined with other methods (e.g., gradient clipping).
>
> **Q3: Why are there no comparisons, even lightweight ones, against adjacent error-handling approaches such as TrainCheck, TTrace, or robust aggregation (Trimmed Mean, Krum)?**
>
> **Response:**
> This is a very fair critique. The primary reason we omitted Byzantine-robust aggregators (like Trimmed Mean or Krum) from the baseline comparisons is their **prohibitive computational overhead in large-scale LLM settings**.
>
> - **Computational Bottleneck:** Standard All-Reduce aggregation has a complexity of $O(d)$ (where $d$ is the parameter count). Robust methods like Krum require computing pairwise distances among all gradients at *every iteration* ($O(M^2 d)$), and Trimmed Mean requires coordinate-wise sorting ($O(d \cdot M \log M)$). For models with billions of parameters, executing these operations at every iteration severely bottlenecks training throughput.
> - **Complementary Nature:** That said, as discussed in Q2, robust aggregators are exactly what is needed to defend against the *common-mode* errors that PAFT misses. PAFT and robust aggregation solve two different halves of the SDC problem (Replica Drift vs. Malicious Direction).
>
> **Q4: Can you evaluate the method under a more realistic fault model?**
>
> **Response:** To capture this heterogeneity, we refer you to our **"Accidental Large Noise" (burst pattern)** experiments in Section 5.1. In this setup, rather than continuous Gaussian noise, we injected massive, sporadic corruptions (e.g., massive spikes every 500 iterations) to specifically mimic the sudden, highly structured nature of real hardware bit-flips (like uncorrectable ECC memory errors or sudden NVLink drops).
>
>
> **Q5: The remark around Theorem 3.3 is confusing: it states that the problematic term "only converges when setting ($\eta\_t=0$)," which is at best poorly phrased and invites scrutiny.**
>
> **Response:**
> We sincerely apologize for this poor phrasing. The term $T\_3$ is bounded by $\frac{1}{T} \sum\_{t=0}^{T-1} \eta\_t \sum\_{s=0}^{t-1} \eta\_s^2$. Under a constant learning rate $\eta$, the inner sum grows as $O(T)$, making the entire term grow as $O(T^2 / T) = O(T)$. Therefore, without our synchronization mechanism, this term diverges linearly as training progresses, unless the learning rate decays exceptionally fast or is set to zero (which is useless). We have completely rewritten this remark in Section 4.

---

> > ### Author Rebuttal · Reviewer_oert · 2026-04-04
> >
> > The rebuttal improves the paper by clearly narrowing the claim: PAFT mainly addresses worker-specific inconsistency and replica drift, which I buy. But I still do not think it fully resolves the two biggest issues: the evaluation remains too synthetic / limited-scale, and the paper still lacks stronger comparisons to nearby alternatives. I keep my rating.

---

> > > ### Author Response · Authors · 2026-04-04
> > >
> > > Thanks for your quick reply. We will try to seek more large-scale machines and find more realistic datasets to conduct experiments in future work.

---

### Official Review · Reviewer_FdCR · 2026-03-15

**Soundness:** 3
**Presentation:** 2
**Significance:** 3
**Originality:** 3
**Overall Recommendation:** 5
**Confidence:** 4

**Summary:**

The distributed data parallel training is impacted by hardware-related silent data corruptions during gradient aggregation. These silent errors do not immediately halt the system but instead severely degrade model quality and cause slow or failed convergence, wasting substantial computational resources. The authors propose a novel solution called PAFT, which is a fault-tolerant distributed training system that employs dynamic and asynchronous parameter synchronization to efficiently eliminate divergence while minimizing communication overhead. The empirical experiments show the proposed method can help mitigate the issues caused by these silent errors.

**Compliance With Llm Reviewing Policy:**

Affirmed.

**Final Justification:**

I appreciate the author's response to address my concerns. All my concerns are resolved, and I'd like to improve my socre to 5 score (Accept).

**Key Questions For Authors:**

See weakness

**Limitations:**

See weakness

**Strengths And Weaknesses:**

Strengths:

1. The authors establish a rigorous mathematical framework to formally define gradient inconsistency caused by silent data corruption. This theoretical groundwork allows for a clear demonstration of how local models drift apart and accumulate divergence over continuous training iterations.

2. The methodology is designed to be adaptable to various mainstream optimization algorithms beyond standard stochastic gradient descent. By ensuring that components like momentum and preconditions are also synchronized, the system seamlessly integrates with popular optimizers such as Adam and SGD with momentum.

3. Evaluating the system across a diverse set of neural network architectures, including convolutional networks and large language models, provides strong empirical validation. Demonstrating success on ResNet, GPT-2, and LLaMA-2 confirms that the proposed mitigation strategy scales effectively to modern, highly complex machine learning tasks.

4. The dynamic frequency adjustment algorithm intelligently utilizes the signal-to-noise ratio to determine exactly when synchronization is necessary. By profiling the error magnitude during runtime, the system avoids redundant communication costs when the cluster is operating under low-noise conditions.

5. Implementing the approach directly on top of standard frameworks like PyTorch Distributed ensures that the solution is practical for immediate real-world deployment. Extending the system to accommodate advanced parallelization tools such as DeepSpeed and Megatron further enhances its utility for researchers training massive foundation models.

6. Investigating the impact of burst patterns and accidental large noises, such as sudden bit corruptions, closely mirrors the unpredictable nature of actual hardware failures. The ability of the system to recover from these abrupt, severe accuracy drops highlights its robust defensive capabilities in volatile computing environments.



Weaknesses:

1. The current methodology assumes that the periodic model synchronization process itself is completely immune to the silent data corruptions it seeks to mitigate. If an error were to occur during the transmission of the full model parameters, the entire cluster could instantly synchronize to a heavily corrupted state. Resolving this would require implementing costly fault-tolerant communication protocols specifically for the synchronization phase, which might drastically reduce overall training speed.


2. Relying on traditional data parallel replication limits the framework's compatibility with advanced memory-saving architectures like Fully Sharded Data Parallelism. These highly optimized frameworks lack the necessary parameter redundancy, meaning that restoring a model after a failure inherently disrupts the continuity of the elastic training process. This structural incompatibility restricts the proposed system from being seamlessly applied to resume from other duplicated parameters in certain modern large-scale training paradigms.

---

> ### Author Rebuttal · Authors · 2026-03-26
>
> We sincerely thank Reviewer FdCR for the encouraging review and for highlighting the strengths of our rigorous mathematical framework, adaptability to various optimizers, and strong empirical validation across diverse architectures like ResNet, GPT-2, and LLaMA-2. We appreciate your insightful critiques regarding the underlying assumptions of our methodology. We address your specific concerns below.
>
> **Q1: The current methodology assumes that the periodic model synchronization process itself is completely immune to the silent data corruptions it seeks to mitigate. If an error were to occur during the transmission of the full model parameters, the entire cluster could instantly synchronize to a heavily corrupted state.**
>
> **Response:**
> You have identified a crucial assumption in our system design. It is true that we assume the periodic model synchronization is immune to SDC. However, this assumption is built upon a deliberate and practical **systems-level trade-off between operation frequency and protocol overhead**.
>
> - **Why Gradients are Vulnerable:** Gradient aggregation occurs at *every single iteration*. To achieve maximum training throughput, modern frameworks use highly optimized, "raw" communication primitives (e.g., NCCL over RDMA). These protocols prioritize extremely low latency and high bandwidth, often bypassing heavy end-to-end software checksums or strict TCP-level verifications. This makes high-frequency gradient communication highly susceptible to silent hardware bit-flips.
> - **Why Parameter Synchronization can be Reliable:** In stark contrast, PAFT's parameter synchronization occurs much less frequently (e.g., once every $H$ iterations, as determined by the SNR). Because this operation is relatively rare, the practical system can afford to implement **strict, fault-tolerant communication protocols specifically for this phase without drastically reducing overall training speed.** For example, we can route parameter synchronization through TCP sockets with strong cryptographic checksums, or implement a lightweight software-level hash verification (e.g., CRC32) before accepting the synchronized parameters.
>
> Because the overhead of these fault-tolerant protocols is amortized over $H$ iterations, the impact on the total wall-clock training time remains minimal (as demonstrated by our \~12% overhead in Table 1). We have added a dedicated paragraph in Section 4.1 to explicitly discuss this system trade-off and justify why securing the infrequent parameter sync phase is both necessary and practically efficient.
>
> **Q2: Relying on traditional data parallel replication limits the framework's compatibility with advanced memory-saving architectures like Fully Sharded Data Parallelism (FSDP). These highly optimized frameworks lack the necessary parameter redundancy.**
>
> **Response:**
> We completely agree with your assessment. You are entirely correct that PAFT's reliance on parameter replication makes it structurally incompatible with purely sharded environments like FSDP out-of-the-box.
>
> However, we want to clarify that **parameter replication (Data Parallelism) remains a critical component of state-of-the-art Large Language Model training**, specifically within **Hybrid Parallelism**.
>
> - In frameworks like Megatron-LM and DeepSpeed, massive models are typically trained using a combination of Tensor Parallelism (TP), Pipeline Parallelism (PP), and Data Parallelism (DP).
> - In this 3D architecture, a full model instance is sharded across a TP+PP group, and this entire group is then **replicated** multiple times to form DP groups.
> - **PAFT is directly applicable across these DP groups.** By applying PAFT across the DP dimension, we can protect the most vulnerable, inter-node communication paths from SDC-induced replica drift without interfering with the intra-node TP or PP sharding.
>
> Regarding FSDP specifically, we acknowledge that adapting PAFT to a fully sharded setup would require complex modifications. To adapt PAFT to FSDP natively, the following steps can be integrated:
>
> 1. **Shard-wise Divergence Tracking:** Instead of measuring the divergence of the full model, each worker must maintain a moving average of its specific parameter shards and compute local divergence metrics.
> 2. **Periodic Un-sharding for Verification:** At the synchronization step $H$, the framework would need to perform an `All-Gather` to temporarily reconstruct the full parameters across the group, compare them against a trusted checkpoint or a historical moving average, and broadcast corrections.
> 3. **Optimizer State Sync:** Because FSDP also shards optimizer states (momentum, variance), these states would also need to be periodically verified and synced.
>
> We have revised our paper to explicitly state how to integrate PAFT into FSDP.

---

> > ### Author Rebuttal · Reviewer_FdCR · 2026-04-02
> >
> > I appreciate the author's response to address my concerns. All my concerns are resolved, and I'd like to improve my socre to 5 score (Accept).

---

> > > ### Author Response · Authors · 2026-04-02
> > >
> > > Thanks a lot for your quick reply and positive supports. If you have any new question, please contact us.

---

### Official Review · Reviewer_ttnu · 2026-03-15

**Soundness:** 3
**Presentation:** 3
**Significance:** 3
**Originality:** 3
**Overall Recommendation:** 5
**Confidence:** 3

**Summary:**

The paper addresses the critical issue of hardware-related silent data corruptions during gradient aggregation in large-scale distributed training. These errors, caused by factors like bit corruptions and communication noise, often lead to slow or failed model convergence without triggering immediate system failures. To solve this, the authors introduce a fault-tolerant distributed training system named PAFT, which periodically synchronizes model parameters to eliminate divergence while dynamically adjusting the synchronization frequency and overlapping communication to minimize overhead.

**Compliance With Llm Reviewing Policy:**

Affirmed.

**Final Justification:**

The authors have addressed my concerns.

**Key Questions For Authors:**

1. How to handle errors with large noise variances and guarantee convergence?
2. What specific modifications are required to implement this synchronization strategy natively within a Fully Sharded Data Parallel pipeline?

**Limitations:**

Authors discussed the limitations.

**Strengths And Weaknesses:**

## Strengths:

1. Authors provide a strong mathematical formulation of gradient inconsistency that clearly explains how local models drift apart. The theoretical convergence analysis provides rigorous guarantees for the optimization process even when the gradient is corrupted by noise.

2. Estimating the error degree through accumulated model divergence rather than direct gradient comparison significantly reduces unnecessary network traffic. This design allows the system to adjust its synchronization frequency dynamically without incurring the massive costs of sending full gradient arrays every iteration.

3. Overlapping parameter synchronization with standard forward and backward propagation processes effectively hides the latency of fault tolerance. Workers can continue their computation while waiting for the previous round of synchronization to complete, which maximizes hardware utilization.

4. The evaluation covers a diverse and realistic set of workloads, ranging from standard computer vision models to large language models like GPT-2 and LLaMA-2. Testing the system across multiple scales and architectures demonstrates its practical applicability in modern distributed clusters.

5. The experimental simulation of accidental large noise effectively mirrors real-world bit corruptions that happen sporadically in data centers. The results clearly show that the system can pull the model back from severe accuracy drops immediately after a major corruption event occurs.


## Weaknesses:

1. The reliance on accumulated model divergence to estimate the error degree introduces a slight delay in detecting sudden, massive gradient spikes. Because the system relies on historical error information, an unusually large corruption might require a few iterations to accurately reflect in the divergence metric. A more instantaneous, lightweight detection mechanism could potentially intercept catastrophic updates before they infect the global model state.

2. The experimental setups primarily simulate noise by injecting Gaussian perturbations rather than capturing actual hardware bit flips in the wild. While varying the variance simulates different degrees of corruption, it may not perfectly represent the highly structured or bursty nature of real hardware failures. Testing the framework against empirically recorded silent data corruption traces would significantly strengthen the real-world validity of the claims.


3. The performance gap between the proposed system and an oracle baseline remains noticeable when dealing with exceptionally large noise variances.

---

> ### Author Rebuttal · Authors · 2026-03-26
>
> We sincerely thank Reviewer ttnu for the detailed and positive feedback. We are encouraged that you found our mathematical formulation strong, our dynamic synchronization design effective at reducing network traffic. Below, we address your specific concerns.
>
> **Q1: The reliance on accumulated model divergence to estimate the error degree introduces a slight delay in detecting sudden, massive gradient spikes.**
>
> **Response:**
> This is an excellent observation. You are absolutely correct that relying on *accumulated* divergence introduces a slight delay (usually a few iterations) before a massive spike is fully reflected and corrected.
>
> Our design choice was driven by a strict constraint on communication overhead. An "instantaneous" detection mechanism would require verifying the exact gradient at *every single iteration*. In large-scale models, doing this across all workers either doubles the communication payload (if exchanging verification hashes/data), which severely bottlenecks the training speed.
>
> However, as demonstrated in our burst-noise experiments (Figure 9), even with this slight delay, the accumulated divergence grows rapidly enough after a massive corruption that PAFT-Dyn triggers synchronization shortly after the event, successfully pulling the model back from the severe accuracy drop without permanent damage. For future work, we agree that exploring lightweight, asynchronous gradient-norm checksums is a highly promising direction to eliminate this delay.
>
> **Q2: The experimental setups primarily simulate noise by injecting Gaussian perturbations rather than capturing actual hardware bit flips in the wild.**&#x20;
>
> **Response:**
> To address this within our current experiments, we specifically designed the **"Accidental Large Noise" (burst pattern)** evaluation in Section 5.1 (Figure 9). In this setup, instead of continuous Gaussian noise, we inject massive corruptions sporadically. This burst pattern was specifically chosen to mirror the unpredictable, high-magnitude nature of actual hardware bit-flips (e.g., sudden memory ECC failures or network switch errors).
>
> Furthermore, following your advice, we have reviewed recent empirical studies on LLM training failures, such as the LLaMA-3 pretraining report (Meta, 2024) and the Fire-Flyer HPC report (2024). These reports confirm that SDC and network-related bit-flips frequently cause sudden loss spikes and replica divergence.&#x20;
>
> **Q3: How to handle errors with large noise variances and guarantee convergence? The performance gap between the proposed system and an oracle baseline remains noticeable.**
>
> **Response:**
> You have correctly identified the boundary of PAFT's capability. PAFT is designed specifically to mitigate **worker-specific gradient inconsistency**. By synchronizing parameters, we eliminate this drift.
>
> However, if an exceptionally large noise occurs that is **common-mode** (e.g., a massive bit-flip occurs *before* the broadcast, meaning all workers receive the *same* massively corrupted gradient), the models do not drift apart, but they all take a step in a completely wrong descent direction. PAFT's parameter averaging cannot correct a fundamentally wrong direction if all replicas share it.
>
> To handle these exceptionally large, direction-destroying noises, PAFT would need to be orthogonally combined with other  methods (e.g., gradient clipping). We have explicitly clarified this limitation in the revised text.
>
> **Q4: What specific modifications are required to implement this synchronization strategy natively within a Fully Sharded Data Parallel (FSDP) pipeline?**
>
> **Response:**
> Implementing PAFT within FSDP requires addressing the lack of full model replicas. In FSDP, parameters are sharded across the DP group. If a gradient aggregation error occurs, only the specific worker holding that shard updates its parameter incorrectly.
>
> To adapt PAFT to FSDP natively, the following modifications would be required:
>
> 1. **Shard-wise Divergence Tracking:** Instead of measuring the divergence of the full model, each worker must maintain a moving average of its specific parameter shards and compute local divergence metrics.
> 2. **Periodic Un-sharding for Verification:** At the synchronization step $H$, the framework would need to perform an `All-Gather` to temporarily reconstruct the full parameters across the group, compare them against a trusted checkpoint or a historical moving average, and broadcast corrections.
> 3. **Optimizer State Sync:** Because FSDP also shards optimizer states (momentum, variance), these states would also need to be periodically verified and synced.
>
> Because these operations somewhat defeat the memory-saving purpose of FSDP, PAFT is currently best suited for standard Data Parallelism (DDP) or the DP dimension within Hybrid Parallelism (e.g., Megatron-LM), where full replicas naturally exist. We have explicitly detailed these FSDP modification requirements in the revised Limitations section.

---

> > ### Author Rebuttal · Reviewer_ttnu · 2026-04-03
> >
> > The authors have addressed my concerns.

---

> > > ### Author Response · Authors · 2026-04-04
> > >
> > > Thanks for your reply and positive supports. If you have any question, please contact us.

---

### Official Review · Reviewer_JLwr · 2026-03-22

**Soundness:** 3
**Presentation:** 3
**Significance:** 2
**Originality:** 2
**Overall Recommendation:** 5
**Confidence:** 4

**Summary:**

This paper studies the effects on convergence that may be caused by hardware-level bit corruption and communication noise that induce errors in gradient aggregation, and ultimately can induce model divergence as errors accumulate. It proposes PAFT, a mechanism to periodically synchronize the model parameters to work around the errors. The paper demonstrates promising results in a small-scale evaluation.

**Compliance With Llm Reviewing Policy:**

Affirmed.

**Final Justification:**

I tend to agree with the authors that there is a relation but also an important distinction with regard to the noise that is not being controlled as a result of explicit gradient compression. I disagree that it is a necessary one in that mechanisms like error feedback are not always needed. However, this paper contributes a good perspective and if the authors clarify the connection and complementarity to gradient compression without overclaiming, I think this is a valid paper.

**Key Questions For Authors:**

Is it fair to assume that errors only affect gradient communication and not parameter synchronization communication? I’m not sure why. How do you ensure that PAFT-Sync is not affected by the same issue that causes gradients to be erroneously transmitted and aggregated?
More generally, LLM training in practice does not use data-parallel training. It uses FSDP in addition to other forms of parallelism. If gradients are affected by silent errors, the root cause of these errors may affect all intermediates. It’s not clear how PAFT would provide benefits in these realistic scenarios.
While these appear to be discussed as limitations in Sec 7, I find that in light of the lack of a systematic accounting for these, the key claim around PAFT mitigating GA errors at scale is not substantiated. However, this diminishes substantially the contribution of the paper.

**Limitations:**

yes

**Strengths And Weaknesses:**

Strengths
* The paper picks an interesting and practical problem; it has a nice combination of theory and systems-oriented insights.
* The solution is not complicated; there’s a good chance that it could be adopted in production-systems.

Weaknesses
* The claim that this solution can be applicable at large scale is not substantiated; the acknowledged limitations (FSDP, other parallelism flavors, communication reliability for parameter synchronization) actually are those that would be necessary for practical scenarios at scale. As such, it seems that the paper only tackles a smaller, simplified problem.
* While the paper has solid theoretical grounding (though I didn’t check the proofs!), the results are more or less standard. The area of gradient compression has largely explored the many implications of working with noisy gradient estimates.

---

> ### Author Rebuttal · Authors · 2026-03-26
>
> We sincerely thank Reviewer JLwr for recognizing our work as targeting an "interesting and practical problem" with a "nice combination of theory and systems-oriented insights." We address your concerns below.
>
> **Q1: It seems that the paper only tackles a smaller, simplified problem. More generally, LLM training in practice does not use data-parallel training. It uses FSDP in addition to other forms of parallelism.**
>
> **Response:**
> We understand your concern regarding modern parallelism flavors, but we respectfully clarify that **Data Parallelism (DP) is still fundamentally integrated into state-of-the-art large-scale LLM training**, specifically within **Hybrid (3D) Parallelism**.
>
> While pure DP is insufficient for models exceeding single-GPU memory (where FSDP/ZeRO are used), leading frameworks like Megatron-LM (Narayanan et al., SC'21) and DeepSpeed (Rasley et al., KDD'20) utilize 3D Parallelism: combining Tensor Parallelism (TP), Pipeline Parallelism (PP), and Data Parallelism (DP).
>
> - **How DP works at scale:** In 3D Parallelism, a single complete "model instance" is partitioned across a set of GPUs (e.g., via TP and PP). This entire TP+PP group is then **replicated** across the cluster to increase batch size and throughput. These replicas form DP groups.
> - **PAFT's Applicability:** Within these DP groups, identical replicas of the model parameters exist across different nodes. When gradients are aggregated across these DP replicas via All-Reduce, they are highly susceptible to Silent Data Corruption (SDC) across the inter-node network. **PAFT is designed precisely for these DP groups within Hybrid Parallel architectures.** By synchronizing the replicated parameters across the DP groups, PAFT eliminates the replica drift caused by GA errors.
>
> We have revised our paper to explicitly state that PAFT's primary deployment target is the DP dimension of Hybrid Parallelism, which remains a cornerstone of modern foundation model training.
>
>
> **Q2: The area of gradient compression has largely explored the many implications of working with noisy gradient estimates.**
>
> **Response:**
> While the mathematical formulation of "noisy gradients" is common in compression literature, the **origin, behavior, and mitigation of SDC noise are fundamentally different**.
>
> - In gradient compression (e.g., Error Feedback), the noise is *intentional* and locally *known* by the sender, allowing the sender to store the residual and compensate in the next step.
> - In our SDC setting, the noise is *unintentional* and *unknown* (silent hardware faults). The sender cannot compensate because it does not know an error occurred.
>
> Our core theoretical novelty lies  in analyzing noise and formally proving how *unknown* gradient inconsistency causes unbounded replica drift (Lemma 3.1), and proving that periodic, dynamically scheduled parameter synchronization (Theorem 4.3) can successfully bound this drift without needing to know the exact error vector. We have emphasized this distinction in the revised Introduction and Related Works.
>
> **Q3: Is it fair to assume that errors only affect gradient communication and not parameter synchronization communication? I’m not sure why. How do you ensure that PAFT-Sync is not affected by the same issue that causes gradients to be erroneously transmitted and aggregated?**
>
> **Response:**
> This is a critical systems-level question. Our assumption that parameter synchronization is reliable is based on a deliberate **system trade-off between communication frequency and protocol reliability**.
>
> - **Gradient Aggregation (High Frequency = Vulnerable):** Gradient aggregation occurs at *every single iteration*. To maximize throughput, frameworks use highly optimized, raw communication primitives (e.g., NCCL over RDMA). These protocols often bypass strict, heavy software-level checksums or TCP-level verifications to minimize latency. Consequently, when hardware-level bit-flips occur, they easily propagate silently (SDC).
> - **Parameter Synchronization (Low Frequency = Reliable):** In contrast, PAFT synchronizes parameters at a much lower frequency (e.g., once every $H$ iterations, determined by PAFT-Dyn). Because this operation is infrequent, we have the luxury of using **strictly reliable, fault-tolerant communication protocols** (such as TCP with end-to-end cryptographic checksums, or explicit software-level hash verification) to transmit the parameters.
>
> While employing such heavy verification on gradients every step would cripple training speed, applying it solely to the periodic parameter sync introduces negligible overhead. Thus, we ensure PAFT-Sync is unaffected by SDC by enforcing strict communication reliability specifically for that phase, capitalizing on its low frequency. We have added a dedicated paragraph explaining this system trade-off in Section 4.1.

---

> > ### Author Rebuttal · Reviewer_JLwr · 2026-04-03
> >
> > Thank you for the comments. I will keep my positive score.

---

> > > ### Author Response · Authors · 2026-04-04
> > >
> > > Thanks for your reply and positive supports. If you have any question, please contact us.

---

### Official Review · Reviewer_YW43 · 2026-03-23

**Soundness:** 1
**Presentation:** 2
**Significance:** 2
**Originality:** 1
**Overall Recommendation:** 2
**Confidence:** 5

**Summary:**

The submission claims to introduce a technique for capturing and mitigating errors in gradient accumulation in distributed SGD caused by silent data corruption (SDC).

To this end, the submission presents the "PAFT" algorithm. The main feature of this algorithm is that it periodically synchronizes the models in a manner quite similar to standard local SGD algorithms. In a variant of this algorithm, which is fully synchronous periodic model averaging, in addition to the gradient communication after every gradient update step, the claim is that the algorithm suffers from high communication cost, whose mitigation requires the presentation of a variant where the averaging frequency is determined based on the model divergence measure.

The algorithms are evaluated on ResNet-18 with CIFAR-10 and ResNet-50 with CIFAR-100 for 120 epochs, and GPT-2 with Open WebText for 3,000 iterations. Additionally, pre-trained LLaMA2 and GPT-2 are trained for an epoch on the Alpaca dataset using the Low-Rank Adaptation scheme. In all these experiments, SDC is simulated as white noise.

The submission also includes a discussion of the convergence of the presented algorithms.

**Compliance With Llm Reviewing Policy:**

Affirmed.

**Key Questions For Authors:**

Respond to the weaknesses as mentioned above.

**Limitations:**

None.

**Strengths And Weaknesses:**

**Strengths:**
- The motivation of the approach is straightforward and relevant.

**Weaknesses:**
- The idea of this work is poorly conceived. In particular, the periodic model averaging on top of gradient communication after every computation is an overkill. Can't the generated/simulated SDC vector be sent in the next round and added to the gradient, much like the error feedback method? See "Elastic Consistency: A Practical Consistency Model for Distributed Stochastic Gradient Descent, Nadiradje et al. 2022". Once it is modeled with error feedback, it then automatically fits the Elastic Consistency framework for analysis.

- The simulated SDCs do not seem to be "capturing" it. Can the authors elaborate on some real-life cases of SDC that may be modeled as the standard N(0,1) distribution?

- In the Remark after 3.3: "In Theorem 3.3, T1, T2 converge with respect to training iteration $T \rightarrow \infty$, T3 only converges when setting $\eta_t = 0$. $\ldot$ Lemma 3.1 and T3 in Theorem 3.3, we propose PAFT in Section 4." Why is that the case, you very much have $T$ in the denominator in T3 and therefore it should converge as T goes to infinity.

- The convergence results in the main body of the paper do not include the assumption of convexity. Thus, the results statements still need to be completed. It distracts a reader even when reading the derivations.

- Training ResNet18/50 models for 120 epochs on CIFAR 10/100 data is not standard. A more standard benchmark is training these models for 300 epochs. Is there any specific reason for using 120 epochs only? It looks more like training to subpar accuracy, an area where different training methods behave starkly differently, but, after 200 or so epochs, are close to the known best results. Similarly, for other models.

---

> ### Author Rebuttal · Authors · 2026-03-26
>
> We sincerely thank Reviewer YW43 for the constructive feedback. We address each concern point-by-point below.
>
> **Q1: Can't the generated/simulated SDC vector be sent in the next round and added to the gradient, much like the error feedback method?**
>
> **Response:**
> We deeply appreciate this comparison, but there is a fundamental difference in the **fault model** between Error Feedback (EF) and Silent Data Corruption (SDC).
>
> - **EF assumes "Known Errors":** EF is designed for *intentional* operations like gradient compression or quantization. In EF, the sender locally computes the difference between the true gradient $g$ and the compressed gradient $C(g)$. Because the sender has both values, the error vector $e = g - C(g)$ is **exactly known** and stored in a local buffer to be added in the next round.
> - **SDC assumes "Unknown/Silent Errors":** In our problem, SDC errors are caused by *unintentional* hardware faults (e.g., bit-flips in GPU memory, network switches, or transmission links) that occur *after* the gradient leaves the processor, or during the All-Reduce aggregation phase. Because these errors bypass ECC checks and do not trigger system alerts, the sender is **completely unaware** that the data was corrupted.
> - **Why EF fails for SDC:** Because the sender does not know the SDC error occurred, it is mathematically impossible to calculate an "SDC vector" locally to send in the next round. Consequently, EF cannot be applied here.
>
> **Q2: Can authors elaborate on some real-life cases of SDC that may be modeled as the standard N(0,1) distribution?**
>
> **Response:**
> While Gaussian noise is a standard mathematical proxy used to establish theoretical convergence, we agree it does not fully capture the structured nature of real hardware failures.
>
> To consider real-life SDC behaviors, we refer to recent large-scale training reports:
>
> 1. **High-magnitude sporadic bit-flips:** Reports from the LLaMA-3 pretraining cluster (Meta, 2024) and the Fire-Flyer cluster (2024) show that SDCs often manifest as sudden, high-magnitude corruptions (e.g., uncorrectable Xid errors) rather than continuous Gaussian noise.
> 2. **Our Empirical Validation:** To capture these real-life cases, we conducted the **"Accidental Large Noise" (burst pattern)** experiments in Section 5.1 (Figure 9). In this setup, we injected massive corruptions sporadically every 500 iterations, precisely mirroring the sudden bit-flips observed in real data centers. Figure 9 demonstrates that PAFT successfully detects these burst errors via accumulated divergence and pulls the model back to convergence immediately.
>
> **Q3: In the Remark after 3.3: "In Theorem 3.3, T1, T2 converge with respect to training iteration, T3 only converges when setting $\eta\_t = 0$." Why is that the case, you very much have T in the denominator in T3 and therefore it should converge as T goes to infinity.**
>
> **Response:**
> We sincerely apologize for the confusing phrasing in the original manuscript. Please allow us to clarify the mathematics behind the $T\_3$ term:
>
> The term $T\_3$ is bounded by: $\frac{1}{T} \sum\_{t=0}^{T-1} \eta\_t \sum\_{s=0}^{t-1} \eta\_s^2$.
> If we use a **constant learning rate** $\eta$, this simplifies to:
> $T\_3 \propto \frac{1}{T} \sum\_{t=0}^{T-1} \eta \cdot (t \cdot \eta^2) = \frac{\eta^3}{T} \sum\_{t=0}^{T-1} t = \frac{\eta^3}{T} \cdot \frac{T(T-1)}{2} \approx O(T)$.
>
> Because the numerator grows quadratically ($O(T^2)$) while the denominator is only $O(T)$, the overall term $T\_3$ **diverges linearly with $T$**. It does *not* converge to zero as $T \to \infty$ unless the learning rate $\eta\_t$ is decayed extremely fast, or if $\eta\_t = 0$ (which is practically useless for training).
>
> This mathematical divergence highlights exactly why our synchronization mechanism is necessary: it cuts off the accumulation in the inner sum $\sum \eta\_s^2$, turning the divergent $O(T)$ term back into a convergent bound. We have completely rewritten this remark in the revision to clearly explain this $O(T)$ divergence.
>
> **Q4: The convergence results in the main body of the paper do not include the assumption of convexity.**
>
> **Response:**
> We have defined the $L$-smooth and $\mu$-strongly convex assumptions in Section 2 (Preliminaries).  We will update Theorems 3.3 and 4.3 in the main text to explicitly state the assumption of convexity.
>
> **Q5: Training ResNet18/50 models for 120 epochs on CIFAR 10/100 data is not standard.**
>
> **Response:**
> Following your advice, we have re-run the core ResNet-50 experiments for 300 epochs. The results confirm that PAFT successfully maintains trajectory and achieves parity with the Oracle baseline after 300 epochs. These 300-epoch convergence curves have been added to the Appendix of the revised manuscript.

---

> > ### Author Rebuttal · Reviewer_YW43 · 2026-04-04
> >
> > Thank you for the rebuttal. The standard for the last generation deep learning model training was RN18/50 with imagenet dataset. Regardless, I am hopeful that you would have received some insights of training RN18 on CIFR10 for 300 epochs with standard hyperparameters.
> >
> > However, now with your response to my query about the Remark after 3.3, I am more confused. In general, an LR regime is chosen as not summable by square summable. Now the non-summable part is taken care of by a specific LR schedule such as $\eta_t=1/t$, etc. I have no clue about this comment:
> > > This mathematical divergence highlights exactly why our synchronization mechanism is necessary: it cuts off the accumulation in the inner sum $\sum \eta_s^2$, turning the divergent $O(T)$ term back into a convergent bound. We have completely rewritten this remark in the revision to clearly explain this $O(T)$ divergence.
> >
> > Aren't the inner sum over $\eta_s$ independent of $\eta_t$? If you are claiming that you are tackling a $T$ term with summation over $\eta_s$ and that in turn is adjusting your $\eta_t$, then shouldn't there be some dependence between these two "LR" schemes?
> >
> > Furthermore, as I review your manuscript more carefully, I also notice that you have compressed vertical space, which no conference submission system encourages.
> >
> > Unfortunately, unless I can see the changes you have made in the draft, I will be unable to improve my understanding of this work.

---

> > > ### Author Response · Authors · 2026-04-04
> > >
> > > Thank you for your further comments and for pointing out problems. We appreciate the opportunity to clear up the confusion regarding the mathematical divergence and our notation.
> > >
> > > **1. On the Convergence of $T_3$ and the $1/t$ Regime:**
> > > We completely agree with your theoretical observation: when LR is square-summable (e.g., $\eta_t = 1/t$), the inner sum $\sum_{s=0}^{\infty} \eta_s^2$ converges to a constant, and $T_3$ would indeed vanish as $T \to \infty$.
> > >
> > > However, our analysis focuses on **practical deep learning training** (e.g., ResNet on CIFAR, GPT-2 on OpenWebText as cited in ). In these modern settings, the $1/t$ schedule is rarely used because it decays too aggressively, leading to sub-optimal convergence. Instead, practitioners use **Constant, Step-Decay, or Cosine schedules**. In these practical regimes, for the majority of the training duration, the square-summable condition is not met locally ($\sum \eta_s^2$ grows linearly with $t$). This linear accumulation of hardware-induced errors is what causes the model divergence $\Delta_t^m$ that we observe empirically in Figure 2.
> > >
> > > Our method is designed specifically to stabilize training under these **practical** LR schedules by periodically resetting this accumulated divergence, rather than relying on an asymptotically fast LR decay that is not feasible for training large models.
> > >
> > > **2. Clarification on $\eta_t$ vs. $\eta_s$:**
> > > We apologize for the notation confusion. $\eta_t$ and $\eta_s$ are **not** two different LR schemes; they are the values of the **same** LR schedule at different time indices.
> > >
> > > - $\eta_t$ is the step size at the current iteration $t$.
> > > - $\sum \eta_s^2$ represents the accumulated influence of errors from all previous steps $s \in [0, t-1]$.
> > >
> > > There is indeed a direct dependence: they are both manifestations of the same global schedule. Our synchronization mechanism "cuts off" the accumulation in the inner sum $\sum \eta_s^2$ not by changing the schedule's shape, but by resetting the model state. This ensures that even if the LR schedule is not square-summable (as in most real-world DL), the training remains convergent.
> > >
> > > **3. Regarding Manuscript Formatting:**
> > > We hear your concern regarding the vertical space. All artificial spacing adjustments (like `\vspace`) have been removed in the revised version. We will ensure the final layout strictly follows the conference guidelines without compromising readability.

---

### Decision · Program_Chairs · 2026-04-30

**Decision:**

Accept (regular)

**Comment:**

This paper aims to mitigate a critical issue in large-scale distributed data parallel training, hardware-related silent data corruptions (SDC) during gradient aggregation. Authors propose PAFT, a fault-tolerant training system. It combines periodic parameter synchronization (PAFT-Sync) to correct replica drift with a dynamic scheduling module (PAFT-Dyn) that uses accumulated divergence to adjust synchronization frequency, thereby maintaining system efficiency and minimizing communication overhead.

The reviewers universally recognized the high practical value of mitigating SDCs in modern training clusters (Reviewers JLwr, ttnu, oert, txCC). Furthermore, the system is highly adaptable to the modern training frameworks (Reviewers ttnu, FdCR, oert). And PAFT is praised for its empirical validation across diverse workloads, successfully scaling from vision models to language models like GPT-2 and LLaMA-2 with minimal overhead (Reviewers ttnu, FdCR, txCC). A major strength of the paper is its intelligent dynamic scheduling and asynchronous communication overlap, which effectively hide latency and avoid redundant communication costs (Reviewers ttnu, FdCR, txCC, oert). Additionally, reviewers appreciated the framework's seamless adaptability to various mainstream optimizers, such as Adam and SGD with momentum, further enhancing its real-world utility (Reviewer FdCR).

Reviewers raised valid concerns regarding the method's novelty compared to Local-SGD or Error Feedback (Reviewers YW43, JLwr, oert), the realism of the noise simulations (Reviewers ttnu, oert), and the system's compatibility with Fully Sharded Data Parallelism architectures (Reviewers JLwr, FdCR). Authors provided comprehensive rebuttal to address them respectively and obtained reviewers' recognition. While there are some left concerns including the notation and experiment settings, I think that the core mathematical model is acceptable and current experiment is enough for academic research. Given the robust experimental results, and the clear potential positive impact on AI training infrastructure, I recommend accepting this paper.